# The unequal effects of the health–economy trade-off during the COVID-19 pandemic

Marco Pangallo [1] ✉, Alberto Aleta [2], R. Maria del Rio-Chanona [3],
Anton Pichler [3], David Martín-Corral [4], Matteo Chinazzi [5,6],
François Lafond [7], Marco Ajelli [8], Esteban Moro [4,9], Yamir Moreno [1,2,3],
Alessandro Vespignani [5] & J. Doyne Farmer [7,10]

Despite the global impact of the coronavirus disease 2019 pandemic, the question of whether mandated interventions have similar economic and public health effects as spontaneous behavioural change remains unresolved. Addressing this question, and understanding differential effects across socioeconomic groups, requires building quantitative and fine-grained mechanistic models. Here we introduce a data-driven, granular, agent-based model that simulates epidemic and economic outcomes across industries, occupations and income levels. We validate the model by reproducing key outcomes of the first wave of coronavirus disease 2019 in the New York metropolitan area. The key mechanism coupling the epidemic and economic modules is the reduction in consumption due to fear of infection. In counterfactual experiments, we show that a similar trade-off between epidemic and economic outcomes exists both when individuals change their behaviour due to fear of infection and when non-pharmaceutical interventions are imposed. Low-income workers, who perform in-person occupations in customer-facing industries, face the strongest trade-off.

From the inception of the coronavirus disease 2019 (COVID-19) pandemic, global efforts have focused mostly on curbing the spread of severe acute respiratory syndrome coronavirus 2 (SARS-CoV-2) through the implementation of mandated non-pharmaceutical interventions (NPIs)[1]. These strategies, encompassing the partial or complete closure of non-essential, customer-facing economic activities such as entertainment and dining, and the enforcement of remote work policies, have impacted people differently across socioeconomic groups. In particular, employees in non-essential industries or able to work remotely were less likely to be exposed to the virus, while workers engaged in essential, in-person tasks experienced a higher risk of exposure. Similarly, the

economic effects of mandated NPIs were industry and occupation specific; for instance, lower-income workers who are primarily engaged in customer-facing industries and in-person occupations were more at risk of layoffs during industry shutdowns[2,3].

In parallel with NPIs, the COVID-19 pandemic also triggered behavioural adaptations, with individuals voluntarily minimizing their contacts and reducing their use of customer-facing services due to fear of the disease. However, the effect of these self-imposed behavioural changes, as opposed to NPIs, remains contentious[4–6]. Moreover, it is an open question whether these behavioural changes result in uneven consequences across socioeconomic groups, similarly to NPIs.

[1]CENTAI Institute, Turin, Italy. [2]Institute for Biocomputation and Physics of Complex Systems and Department of Theoretical Physics, University of Zaragoza, Zaragoza, Spain. [3]Complexity Science Hub, Vienna, Austria. [4]Department of Mathematics and GISC, Universidad Carlos III de Madrid, Leganes, Spain. [5]MOBS Lab, Northeastern University, Boston, MA, USA. [6]The Roux Institute, Northeastern University, Portland, ME, USA. [7]Institute for New Economic Thinking at the Oxford Martin School, and Smith School of Enterprise and the Environment, University of Oxford, Oxford, UK. [8]Laboratory for Computational Epidemiology and Public Health, Department of Epidemiology and Biostatistics, Indiana University School of Public Health, Bloomington, IN, USA. [9]Connection Science, Institute for Data Science and Society, MIT, Cambridge, MA, USA. [10]Santa Fe Institute, Santa Fe, NM, USA. ✉e-mail: marco.pangallo@centai.eu

Addressing the effects of mandated NPIs and behavioural changes, both at the aggregate and granular level, requires building theoretical, mechanistic models that jointly simulate epidemic and economic dynamics at a fine-grained level. While several models have been proposed[7–11], they tend to provide aggregate perspectives on either the epidemic or economic dimension, and thus fall short in characterizing heterogeneous outcomes across diverse socioeconomic groups. The few agent-based models (ABMs) that simulate epidemic spreading and economic decisions at the level of individual, heterogeneous agents[12–14] are primarily designed for qualitative assessments of different policies, using a basic parameter calibration that considers only a few aggregate data points.

In this Article, we introduce an ABM developed to simulate the epidemic and economic impacts on a large synthetic population representative of the New York–Newark–Jersey City, NY–NJ–PA metro area. The model incorporates detailed socioeconomic attributes of agents, along with their consumption and contact patterns derived from comprehensive census, survey and mobility data. The structure of the economy is initialized from input–output tables and national and regional accounts. This joint epidemic–economic ABM largely extends our former epidemic[15,16] and economic[2,17] COVID-19 models.

## Results

We build a data-driven, granular ABM of the New York–Newark–Jersey City, NY–NJ–PA metro area. The main agents of the model are the 416,442 individuals of a synthetic population that is representative of the real population across multiple socioeconomic characteristics, including household composition, age, income, occupation and possibility to work from home (WFH) (for a schematic representation, see Fig. 1, and for a detailed description of the model, see Methods and Supplementary Information).

The epidemic module of the ABM is built on the contact network that connects synthetic individuals. This network has multiple layers, where each layer captures interactions occurring (1) in the household, (2) in school, (3) in the workplace and (4) in the community (during on-site consumption, such as in shops, restaurants or movie theatres). Epidemic propagation occurs on these networks, built via anonymized, privacy-enhanced mobility data from opted-in users, which inform workplace and community interactions. These data, collected through a General Data Protection Regulation (GDPR)-compliant framework by Cuebiq, provide daily workplace visitation patterns and estimates for colocation probabilities in community spaces, based on a Foursquare dataset. The ABM employs a stochastic, discrete-time disease transmission model on the contact network and synthetic population, with individuals transitioning between epidemic states based on key time-to-event intervals (for example, incubation period, generation time and so on) derived from SARS-CoV-2 transmission data.

From an economic point of view, individuals play a role both as workers and consumers. They work in one of multiple industries, producing goods and services that are either sold to other industries as intermediate products or sold to final consumers as consumption products. The economic module specifically emphasizes employment and consumption. In particular, hiring and firing decisions are driven by industry workforce requirements, closures of economic activities and possibility of remote work. Consumption patterns vary among agents based on age and income, and dynamically adjust in response to the evolving state of the pandemic. Specifically, households tend to curtail their demand for services from customer-facing industries because of the fear of the disease (customer-facing industries are entertainment, accommodation–food, other services, retail, transportation, health and education; Supplementary Section 3.1.3). The model also considers the input–output network of intermediates that industries use to produce final goods and services[18], leading to the propagation of COVID-19 shocks to the entire economy.

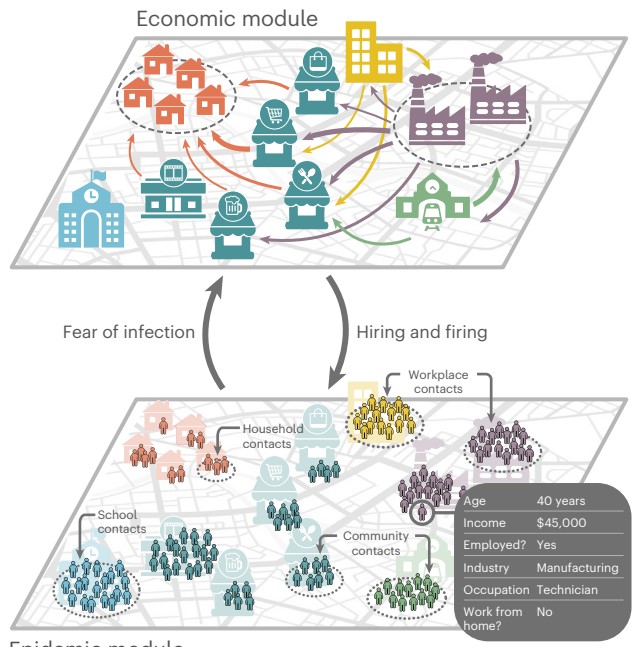

**Fig. 1 | Schematic representation of the joint epidemic–economic model.** The economic module depicts the flow of goods and services between industries and from industries to final consumers (input–output network), while the epidemic module tracks pathogen exposure at workplaces, community/consumption venues, schools and households (contact network). Agents display high heterogeneity across various socioeconomic characteristics (see box). The economic and epidemic modules are closely linked: the epidemic module impacts economic outcomes by reducing consumption due to infection fear, while the economic model influences epidemic spread by altering workplace and community contacts through employment changes in different industries.

### Modelling the first wave of COVID-19 in New York

We calibrate the model's key parameters, including the parameter regulating the behavioural changes, dubbed 'fear of infection', to fit crucial epidemic and economic statistics from the first wave in the NY metro area (Supplementary Section 4). Epidemiological parameters are adjusted to fit ancestral SARS-CoV-2 lineages (Supplementary Table 5). Simulations start on 12 February 2020, protective measures are imposed on 16 March and relaxed on 15 May 15, and simulations end on 30 June. As protective measures, we close schools, mandate WFH and shut down all non-essential economic activities, such as entertainment and most of the accommodation–food industry, but also large parts of manufacturing and construction. We use the official NY regulations to estimate the degree to which a given industry is essential (Supplementary Section 3.2.1) and assume that workers who can WFH are not directly affected by these closures[2]. We name this set of assumptions the empirical scenario.

### Economic validation

Our model accurately matches the six official economic statistics we calibrated it on (Fig. 2a). It correctly reproduces the fact that employment declined more strongly than gross domestic product (GDP) (this is because industries most affected by shutdown orders produce less output per worker). It also correctly reproduces the fact that consumption of goods and services produced by customer-facing industries declined more strongly than consumption of goods and services produced by industries that are not customer facing, ranging from manufacturing products to utilities and financial services (Supplementary Fig. 15).

Our model can also be validated against empirical properties that were not directly targeted in the parameter calibration procedure (Fig. 2c,d). First, thanks to our estimate of pandemic shocks[2] and shock

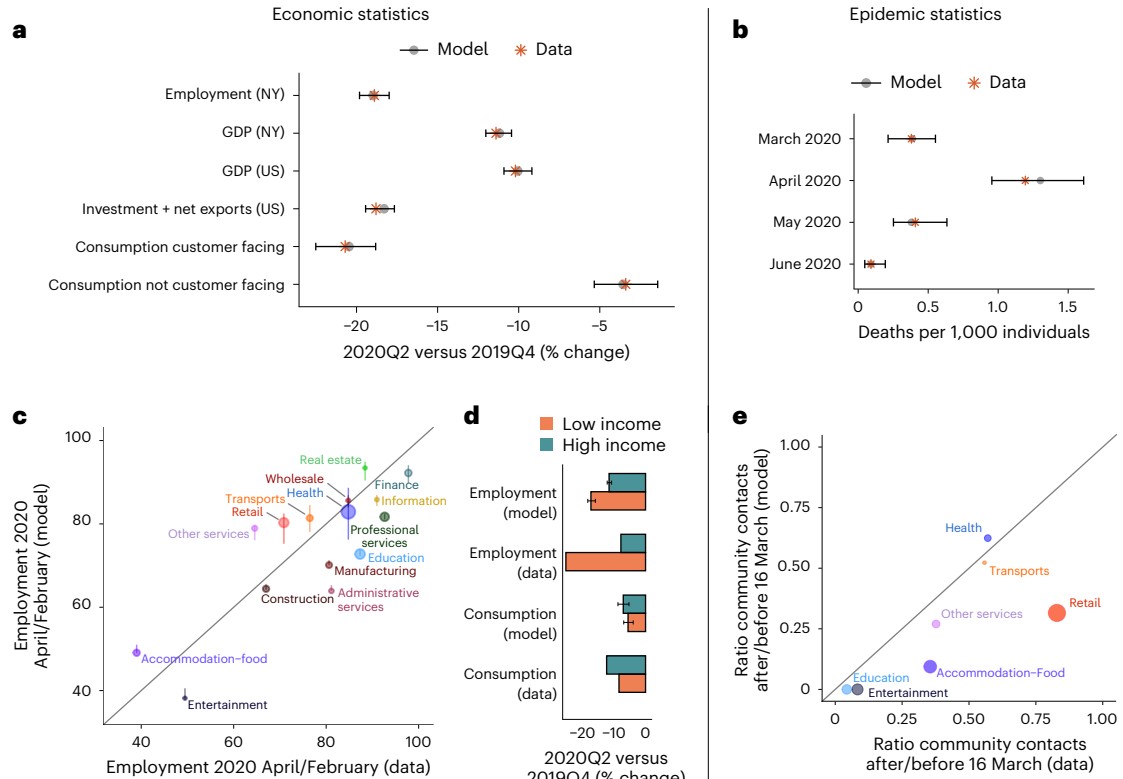

**Fig. 2 | The first wave of COVID-19 in New York: empirical scenario.**
**a**,**b**, Statistics that were directly targeted in the parameter calibration. **a**, The percentage change from October–December 2019 (2019Q4) to April–May 2020 (2020Q2) across six official economic statistics, in the model and in the data (Supplementary Section 4). Here, and throughout the paper, we report the mean and error bars (2.5–97.5 percentile range) across simulation runs that differ by stochastic factors (Methods). **b**, A comparison between model and data for the number of COVID-19 deaths, which is the key epidemic statistic that we targeted. **c**–**e**, Validation results for statistics that were not directly targeted. **c**, Employment in April 2020 as a percentage of employment in February 2020,

across the main two-digit NAICS industries, in the model and in the data. The circle size is proportional to employment in February 2020. **d**, Employment and consumption, in the model and in the data[3], among low-income and high-income households (low income <$27,000 and high income >$60,000; these bands are chosen for comparison to real data; Supplementary Section 5.1.1). **e**, The ratio between community contacts with infectious individuals (Supplementary Section 5.2) before and after the imposition of protective measures, in the model and in the data, for the seven customer-facing industries (Supplementary Section 3.1.3). The circle size is proportional to the share of pre-pandemic contacts.

propagation model[17], we are able to recover industry-specific changes in employment induced by the pandemic (Fig. 2c), with a Pearson correlation coefficient of 0.82 ($P$ value $2 \times 10^{-4}$) between model and data. This accuracy is in large part due to our model's ability to take industry-specific estimates as inputs, but also to the propagation mechanisms embedded in the model. Indeed, the estimates by supply shocks alone have only a Pearson correlation of 0.69 ($P$ value $4 \times 10^{-3}$, Supplementary Fig. 14). Second, thanks to our granular and data-driven characterization of employment and consumption patterns (Supplementary Figs. 8 and 12), we reproduce a key fact: low-income individuals were more likely to become unemployed but reduced consumption less than high-income individuals[3,19] (Fig. 2d). This happens because low-income individuals are more likely to work in the occupations most affected by closures, such as 'food preparation and serving', 'building and grounds cleaning' and 'personal care and service' (Supplementary Fig. 16), but they spend a larger share of their income on essential goods and services such as housing and utilities (we do not consider here the effect of the stimulus programme, which would further increase the spending of low-income individuals).

**Epidemic validation**
On the epidemic side, our model correctly matches the death count data on which it has been calibrated, correctly replicating the spike in the number of reported deaths in April 2020 and the strong reduction in June (Fig. 2d and Supplementary Fig. 18). It also correctly estimates

the changes in contact patterns that occurred after protective measures were implemented, although these data were not used for parameter calibration (Fig. 2e). Both in the model and in the data, community contacts substantially reduced (Pearson 0.75 and $P$ value 0.05), more in mostly non-essential industries such as entertainment and restaurants than in mostly essential industries such as retail and health. We also accurately estimate the reduction in workplace contacts across industries (Supplementary Fig. 20; Pearson 0.88 and $P$ value $5 \times 10^{-6}$), the temporal profile of reduction in contacts (Supplementary Fig. 19) and the increase in prevalence over time (Supplementary Fig. 18). Finally, the model makes a number of estimates about how many infections happen across each layer and industry over time, as well as which occupation, income and age groups are most affected (Supplementary Figs. 21–23). While we are not able to find data to quantitatively evaluate these estimates, our literature review provides some support to these findings (Supplementary Section 5.2.1).

**Counterfactual scenarios**
In our analysis, we quantitatively explore the effects of three key factors that shaped the behavioural and policy response to the first COVID-19 wave. In the following, we use 'baseline' to refer to estimated parameters calibrated with empirical data.

As a first set of counterfactual scenarios, we explore the effect of adjusting the magnitude of the fear of infection parameter that regulates behaviour change. Generally, we treat this parameter as uniform

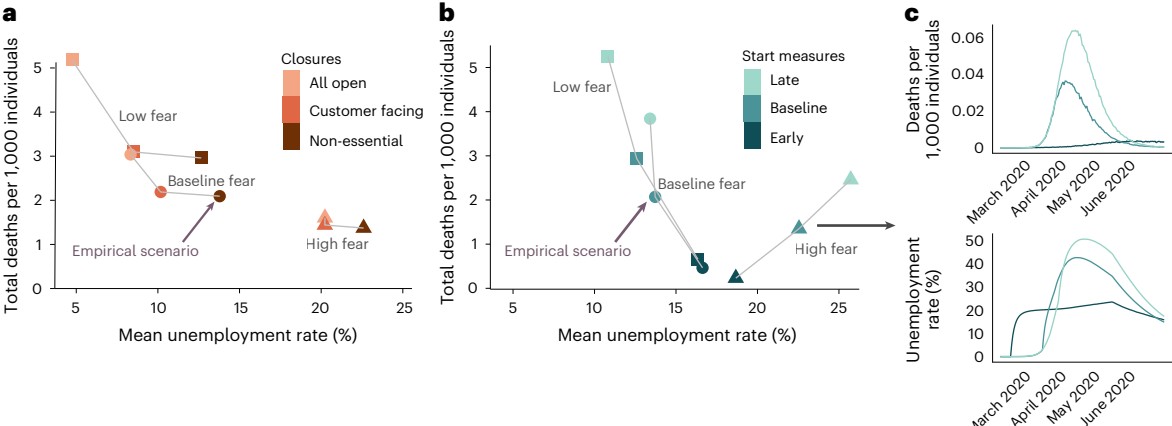

**Fig. 3 | Aggregate results on counterfactuals. a,b**, Deaths and unemployment across scenarios. For each scenario, we show the aggregate unemployment rate and the cumulative number of deaths, as averaged throughout the simulation period and the simulation runs (Supplementary Fig. 26 shows the variability across simulation runs and discusses its interpretation). The empirical scenario is highlighted to serve as a benchmark. Scenarios are distinguished by the strength of behaviour change, as exemplified by the fear of infection parameter (square: low; circle: baseline; triangle: high). **a**, Scenarios are further distinguished by the specific closure of economic activities (all non-essential industries, as occurred empirically, only customer-facing industries and no closures), keeping the start of protective measures fixed at the baseline, empirically observed date. **b**, Scenarios are further distinguished by the start of protective measures (baseline: 16 March 2020, as empirically; early: 17 February 2020 and late: 30 March 2020), keeping closures fixed at all non-essential industries. **c**, For the specific combination of high fear of infection and three different starts of protective measures, a time series of unemployment and deaths corresponding to the three scenarios is shown.

across individuals, per survey evidence[20]. However, we also consider an age-specific fear counterfactual. Our baseline calibration yields a fear of infection parameter distribution (Supplementary Fig. 13), implying a 14% consumption demand reduction in customer-facing industries due to infection fear at the epidemic peak. This calibrated value, which merges NPI effects with behaviour change, cannot be causally interpreted. Absence of NPIs would necessitate a steeper consumption drop for the model to explain observed behavioural changes, thus estimating higher infection fear. But the lack of real-world data without NPIs hampers this estimate. Instead, we explore two counterfactuals: 'low' (0.1 times baseline) and 'high' (10 times baseline), translating to a 1% and 77% consumption reduction due to fear at the peak, respectively. These scenarios facilitate comparing stronger fear of infection effects with stricter NPIs.

Next, we vary two policy-related factors. First, we experiment with different economic activity closures. Besides the baseline scenario with all non-essential industries closed, we consider two milder closure scenarios: (1) only non-essential customer-facing industries are closed and (2) no closures, with all economic activities open. Second, we simulate protective measures starting either 4 weeks earlier (17 February 2020) or 2 weeks later (30 March 2020). Additional counterfactuals, including partial closure of customer-facing industries and no WFH or school closures, are explored in Supplementary Information (Supplementary Figs. 24 and 25).

**Behaviour change versus NPIs**

Aggregate economic and epidemic results are shown in Fig. 3, while results disaggregated by income, geography and industry are shown in Fig. 4 (Supplementary Figs. 27–33). Figure 3a conveys our first main result: stricter closure of economic activities and higher fear of infection both lead to increased unemployment and fewer COVID-19 deaths. To illustrate this, consider a scenario with baseline fear of infection and all economic activities open, represented by the light-coloured circle. If we maintain the fear of infection at the baseline level, but close all non-essential economic activities (as in the baseline scenario), unemployment surges by 64%, while the number of deaths drops by 35%. Likewise, if we instead keep the closure level at the empirical baseline but increase fear of infection (represented by the dark triangle), we see a 40% rise in unemployment and a 50% decrease in deaths relative to the empirical scenario. Similar trends are observed in other scenarios.

Although the total death count and average unemployment can vary substantially across simulation runs, the relative impacts of different policies remain robust (Supplementary Fig. 26).

Both higher fear of infection and stricter closures lead to saving lives at the expense of jobs, for low and high income workers alike (Fig. 4a). However, for low income workers, higher fear of infection or stricter closures have a larger effect, leading to more lives saved and more jobs lost, compared with high-income workers. As we will show later, outside the household setting, most infections occur in customer-facing industries, where most low-income workers are concentrated. Thus, mandated closure or spontaneous avoidance of these industries leads to both more unemployment and fewer workplace infections among low-income workers.

The unequal economic outcomes of the empirical scenario also lead to geographical disparities. Figure 4b shows two maps of unemployment in Manhattan in the empirical scenario (asterisk) and in a counterfactual with low fear and no closures (hash). We see that in the counterfactual, the unemployment rate is very evenly spatially distributed, while in the empirical scenario, low-income areas such as the Queens and the Bronx have a high unemployment rate of more than 20%, compared with high-income areas such as Manhattan, with unemployment rates around 15%.

Overall, these results contribute to the ongoing debate on the relative effectiveness of behavioural change versus NPIs in preserving both public health and the economy. While it is intuitive to expect stricter mandated NPIs to increase unemployment and decrease COVID-19 deaths, it is less apparent that heightened behavioural adaptation would yield similar results (Discussion). Our findings highlight a qualitative parallel between substantial behavioural change and stringent economic activity closures. Spontaneous avoidance of services offered by customer-facing industries, akin to their mandated closure, results in increased unemployment but fewer fatalities. This trend is particularly pronounced among low-income individuals.

**Industry-specific closures**

Our model also evaluates the efficacy of closing all non-essential economic activities, including large segments of manufacturing and construction, compared with exclusively closing customer-facing industries. We find that the mandated closure of all non-essential

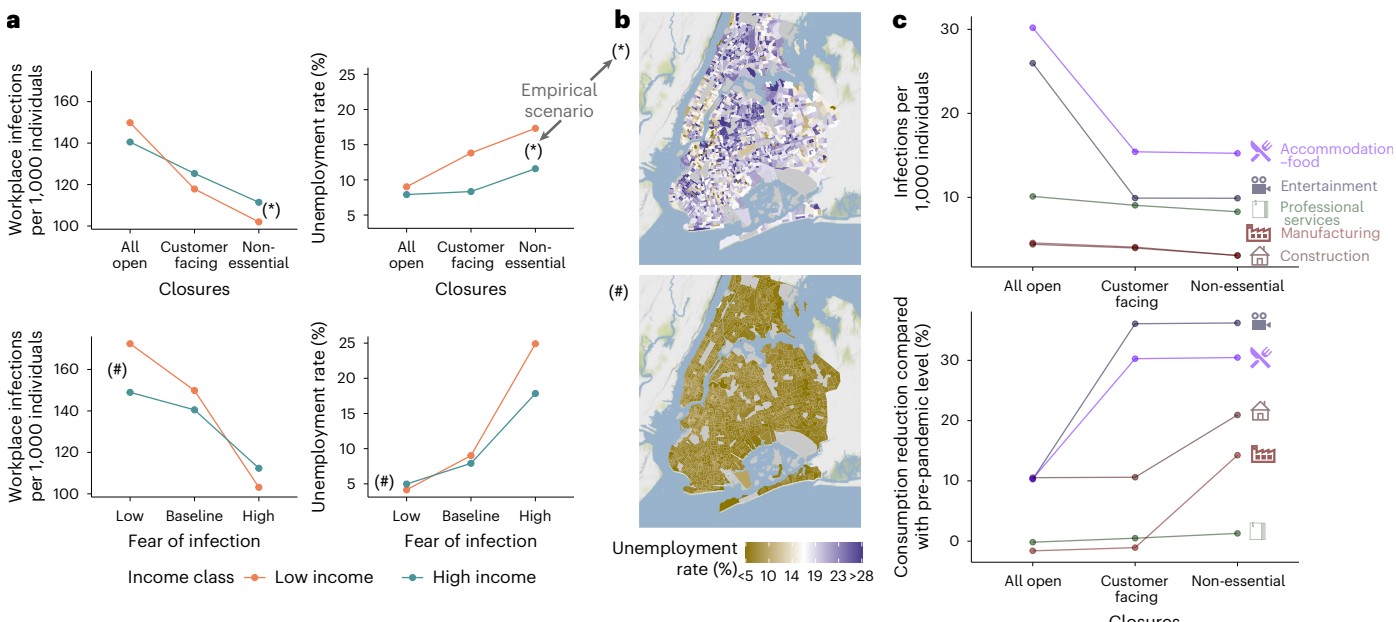

**Fig. 4 | Results on counterfactuals disaggregated by income, geography and industry. a**, Workplace infections and unemployment across income classes. Top: we vary the level of closures, keeping fear of infection and the start of protective measures to their baseline values (so the case with all non-essential activities closed is the empirical scenario). Bottom: we vary fear of infection keeping all economic activities open and starting protective measures on the baseline date. **b**, Maps of unemployment across census tracts in New York City, corresponding to two scenarios in **a**, including the empirical scenario (asterisk) and the counterfactual with no closures and low fear (hash). **c**, Infections and reduction in consumption across five selected industries and three levels of closures, for baseline fear and start of protective measures.

activities only marginally decreases deaths compared with solely closing customer-facing industries, but it drastically increases unemployment. In comparison with the baseline scenario, a counterfactual that only closes customer-facing industries results in a slightly higher death rate (4% higher), but substantially mitigates unemployment, reducing it by 36%. To explain these results, consider Fig. 4c. Most infections occur in customer-facing sectors such as 'entertainment' and 'accommodation–food'. Their closure curbs infections considerably but also consumption and employment. Conversely, closing 'manufacturing' or 'construction' marginally impacts infections but drastically reduces consumption. 'Professional services' remains largely unaffected also because of WFH adaptability.

Methodologically, these industry-specific results were obtained because we associated each consumption venue from mobility data to an economic activity, allowing the quantification of industry-specific contacts. This granular, data-driven approach provides insights that more aggregate, qualitative models might overlook.

### Timing of interventions

Another counterfactual exploration concerns the effectiveness of starting epidemic mitigation and control earlier (4 weeks before) or later (2 weeks later) than in the empirical scenario. As we show in Fig. 3b, delaying these measures marginally reduces unemployment by 2% but causes a notable 50% rise in deaths. In a high fear-of-infection scenario, late measures result in both a 46% increase in deaths and a 12% rise in unemployment. The mechanism for these results is suggested in Fig. 3c, where we show time series across the three counterfactuals with high fear of infection. We see that an early start of mitigation measures prevents an epidemic wave, leading to no further increase in unemployment due to fear of infection. Conversely, with a baseline or late start, substantial behaviour change leads to reduced consumption, and this, in turn, leads industries to fire their employees, increasing unemployment. Thus, starting mitigation measures early is crucial to improve epidemic outcomes, and possibly economic outcomes

too. Our preliminary investigation (Supplementary Section 6.4) also shows that with an early start of protective measures, it is possible to avoid an excessive burden on the healthcare system, as measured by a usage of more than 50% of the nominal capacity of intensive care units.

### Age-specific fear of infection

In the empirical scenario, the parameterization of the 'fear of infection' is uniformly applied across individuals, as suggested by survey evidence[20]. However, we also examined a counterfactual in which young individuals adopt less behavioural changes (low fear of the disease) than older individuals (high fear of the disease), considering this might be a more optimal situation for pandemic control through behavioural change. Here, at-risk older individuals would internalize the infection risk more, while younger individuals, less likely to suffer severe consequences, could maintain higher consumption and contribute to herd immunity. We explore how this scenario plays out quantitatively in the data-driven, granular agent-based model.

To explore the effects of heterogeneous behavioural changes, we group all households into three classes based on the age of their household head (0–34, 35–64 and 65+ years). We assume that fear of infection in each class is proportional to the risk of death in that class (Supplementary Section 6.5). We also normalize the fear of infection parameter across age bracket so that the mean takes the same baseline value that we considered in the empirical scenario. This normalization ensures that results are driven by a different distribution of fear across age groups, rather than by changes in overall fear. At the end of this procedure, the fear level among households aged 0–34 years is 0.02 times the baseline, households aged 35–64 years have a fear level of 0.48 times the baseline and households aged 65+ years have a considerably higher fear level of 4.91 times the baseline. We compare the age-specific scenario with the scenario in which all agents have uniform baseline fear (as in the empirical scenario).

We report aggregate results in Fig. 5a, which reproduces the same scenarios as Fig. 3 for uniform fear, next to the new results for

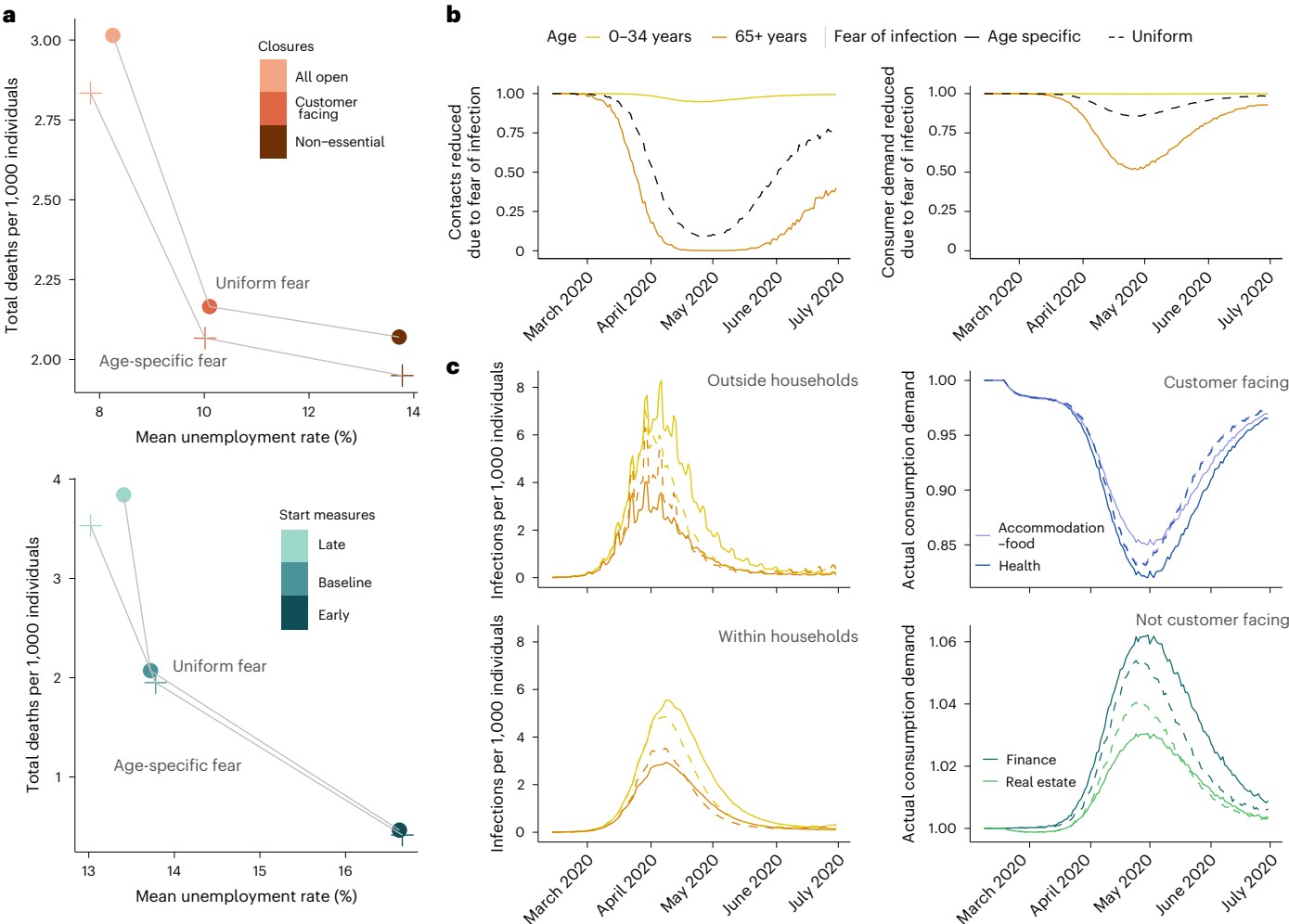

**Fig. 5 | Results on age-specific fear of infection. a**, Aggregate deaths and unemployment across scenarios. The general interpretation is the same as Fig. 3; here, uniform fear is represented by circles and age-specific fear is represented by plus symbols. **b**, For the scenario 'all open–early start', time series of the level of workplace and community contacts and consumption demand of customer-facing industries, disaggregated by type of fear (solid lines, age-specific fear and dashed lines, uniform fear) and by age groups with heterogeneous fear.

These time series show how fear of infection reduces contacts and consumption demand. **c**, For the same scenario as **b**, the time series of infections per 1,000 individuals disaggregated by age groups and by whether they occurred outside or within households, and actual consumption demand (relative to the pre-pandemic level) disaggregated by industry and by whether industries are customer facing or not.

age-specific fear. Adjusting for age-specific fear, while keeping other factors constant, marginally reduces both unemployment and deaths compared with uniform fear. In the 'all open' scenario, where the effect is most pronounced, age-specific fear reduces deaths by 6% and unemployment by 5%. For comparison, closing customer-facing industries cuts deaths by 28% but increases unemployment by 22%.

In Fig. 5b,c, we examine industry- and age-specific effects, focusing on ages 0–34 years and 65+ years. First, in Fig. 5b, we show how fear of infection reduces contacts and consumption demand. As expected, uniform fear leads to equal reductions across all ages by construction, while age-specific fear instead leads to the least reduction in young agents and the most in older agents. Total consumption decreases less than contacts as it may not require direct contact, such as ordering takeaway food (Methods).

In Fig. 5c, we first consider infections (left plots). We distinguish infections occurring inside households from those happening outside (community or workplace contacts). Outside the household, infections among older agents decrease with age-specific fear compared with uniform fear, especially around the epidemic peak where they are 30% lower. However, in the waning phase of the epidemic, infections are

comparable in both scenarios. In contrast, we see an opposite trend among young households, where a large number of infections happen later due to their very low fear of infection. Within households, the differences between age-specific and uniform scenarios are less pronounced.

The relatively small decrease in deaths with age-specific fear can be explained in two ways. On the one hand, the time series of reductions in contacts and infections show that older individuals drastically reduce their contacts only after the epidemic peak. This delay results from the lag between infection and death reporting; behaviour change intensifies when individuals become aware of the number of COVID-19 deaths. On the other hand, older individuals cannot avoid infections within their own households.

As we can also see in Fig. 5c (right plots), age-dependent fear of infection alters consumption demand across industries. Consumption demand decreases more in health, a customer-facing industry on which old agents spend a disproportionate amount of income, and less in accommodation–food, on which young agents spend a higher share of their income. At the same time, consumption demand increases towards industries that are not customer facing because households

reallocate part of their consumption budget to those industries. As individuals in older age groups decrease consumption more and thus have more budget to reallocate because of higher fear of infection, this results in higher consumption demand towards industries that have high consumption share among old individuals, such as finance. By contrast, because younger individuals spend a large fraction of their income on real estate and, with low fear, they do not reallocate much, the increase in demand for real estate services is lower than if fear of infection was uniform across ages.

In summary, these findings show that even when individuals adjust their behaviours in response to their personal risk levels during a pandemic, it only modestly affects health and economic outcomes. Moreover, our results quantify the complex ripple effects across various sociodemographic groups.

## Discussion

Addressing the health impacts of the COVID-19 pandemic required important societal and economic disruptions, sparking intense debates. On the one hand, stringent restrictions and government-enforced measures were critical to suppress the virus spread. On the other hand, some contend that individual behavioural adaptations could have served as a more effective tool in managing the epidemic's trajectory. They suggest that allowing individuals to spontaneously lower their exposure risk according to the epidemic trajectory would lead to the most favourable balance of health and economic outcomes.

Determining whether behavioural change or NPIs are more effective in minimizing the pandemic's health and economic impacts is complex. Each operates differently; NPIs function by curtailing labour supply, creating a supply shock, while behavioural changes act as a demand shock with time-varying effects. Additionally, behavioural changes typically occur only when reported deaths rise considerably, which usually lags about 3 weeks behind infection transmission. The effectiveness of behavioural changes versus NPIs depends on quantitative details such as how long they take to reduce virus circulation to very low levels, enabling a prompt rebound in consumption.

Our findings indicate a parallel between behavioural responses and economic activity shutdowns: both substantial behavioural changes and stringent closures lead to similar patterns of rising unemployment and fewer infections. This impact is particularly heightened among low-income workers compared with high-income workers. Furthermore, this trend persists even when older individuals demonstrate stronger behavioural responses than younger individuals. Indeed, even if individuals change their behaviour proportionally to their own age-specific death risk, it only slightly enhances epidemic and economic outcomes compared with a situation where behavioural change is uniformly distributed across the population. Our results also show that the trade-off between health and the economy strongly depends on which economic activities are closed. The closure of non-customer-facing industries, such as manufacturing and construction, results in a substantial spike in unemployment with only a marginal decrease in fatalities. Additionally, implementing protective measures late in high fear-of-infection scenarios leads to a dual blow of increased deaths and unemployment. In other cases, a delayed start of protective measures substantially escalates fatalities while only slightly reducing unemployment. These results underscore a crucial distinction between behavioural changes and NPIs: while behavioural changes are a result of self-organization, NPIs can be implemented as soon as needed for highest effectiveness.

Our results have the usual limitations pertaining to modelling studies (for more details on the model, see Methods and Fig. 6). In this paper, we exclusively focus on the first wave of COVID-19 in one specific metropolitan area. It could also be important to consider other aspects of the COVID-19 pandemic that became relevant after the first wave of infections such as masks, test, trace and quarantining, variants, vaccination and waning of immunity. However, we expect our key results to hold, and we view our model as mostly applicable to the short-term management of emerging/re-emerging infectious diseases. Another important limitation is that the matching between synthetic individuals and mobility traces is probabilistic, as we do not have socioeconomic information about specific Cuebiq users. Nonetheless, our privacy-preserving matching algorithm based on census tracts is likely to be accurate given the strong socioeconomic disparities in different parts of the New York metro area. From the epidemiological standpoint, we assume the same per-contact risk of infection in different occupational settings. If empirical epidemiological data were collected about the contribution of these settings to SARS-CoV-2 transmission, our estimates could be further refined. Moreover, we did not consider differential risk of severe disease and death for individuals with different socioeconomic status. The inclusion of this factor into the model could further exacerbate the highlighted heterogeneity in the health and economic impact of the pandemic and adopted policies on different segments of the population. From the economic standpoint, we consider industries located over the entire metro area, rather than heterogeneous firms at specific geographical locations. Although this is a limitation of our analysis, we believe that this is the right level of aggregation for the questions considered here, but we acknowledge that a more detailed representation of the production sector may be needed to address questions such as the effectiveness of spatially targeted lockdowns. Finally, the infection transmission and economic models are combined through a 'fear' mechanism that was modelled in a simple manner (that is, as a function of the number of reported deaths on the previous day). For example, individuals may not retrieve information on a daily basis and media may amplify some information (for example, a spike in the number of deaths) at certain times, thus altering the perception of the population[21]; different segments of the population may have a different risk perception[20]. This highlights the importance of conducting future studies to better characterize the relation between risk perception and human behaviour during epidemic outbreaks.

From a policy standpoint, we focus on strategies actually implemented in the real world. We also performed preliminary explorations of more sophisticated policy options, including the activation of protective measures when infections go past a certain threshold, and the deactivation when they go below another threshold (Supplementary Section 6.6). Exploring these strategies, we find interesting results, such as the possibility of quasi-steady states with intermittent closures, but these results do not change our main conclusions. While our model can support policymakers in exploring these scenarios, assessing their practical feasibility is essential and requires case-by-case examination, based on resources, logistics and the objective functions to optimize, which is beyond this paper's scope. Our findings, however, underscore the importance of some targeted policies. For instance, closing customer-facing industries, particularly if done early, effectively reduces viral transmission. This allows income-support schemes to specifically aid certain occupations such as food preparation, serving or personal care services, rather than a broader worker base such as those in construction or manufacturing. Enhanced surveillance and contact tracing in industries employing these low-income workers could yield both health and economic benefits. Importantly, these policies are crucial not only during government-mandated closures but also when spontaneous behaviour changes reduce consumption in these industries. Our findings can thus guide the development of policies to mitigate the health and economic impacts of pandemics, while also safeguarding low-income populations to reduce inequalities.

The model presented in this paper has potential impact on both epidemiological and economic impact analysis. From an epidemiological perspective, we have incorporated industries, occupations and the feasibility of remote work into a granular transmission model, demonstrating the value of integrating economic, social and behavioural dimensions into epidemic spreading models[22,23]. Economic impact

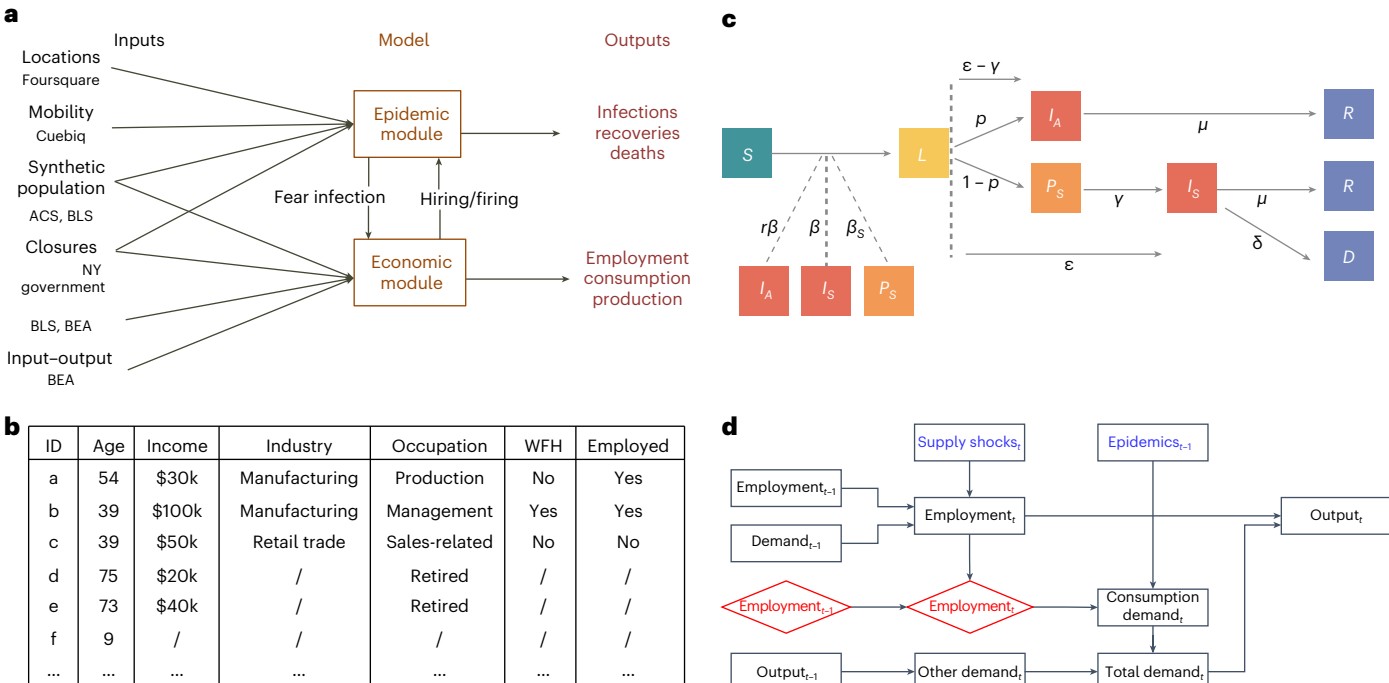

**Fig. 6 | A schematic of the methodology. a**, Inputs and outputs of the epidemic and economic modules. For each group of input data, we list the data source. The feedback from the epidemic to the economic module is through fear of infection, while a reverse feedback occurs through hiring and firing. **b**, Socioeconomic characteristics of six example synthetic individuals, including age, income, industry, occupation, possibility to WFH and employment status. **c**, A schematic of the epidemic module. The different compartments and transition times from one compartment to another are shown. See the text for the meaning of the symbols. **d**, A causal diagram of the economic module. Blue rectangles show processes exogenous to the economic module, black rectangles represent industry-level endogenous variables and red diamonds are individual-level endogenous variables (in this case, the employment status of each individual). A link between variables indicates that the linking variable influences the linked variable. We only show the key variables. On the left, we show lagged values of these variables at time $t-1$, while on the right we show the relations between these variables at time $t$.

studies of disasters frequently limit their focus to industries[24,25], or employ an aggregate representative household[17]. The detailed synthetic population used in our model, mirroring the real-world population, enables us to explore a new array of questions centred around understanding the disparate impacts across socioeconomic groups. In doing so, this paper has potential impact in future research aimed at formulating more targeted, effective public health policies and strategies.

## Methods

This section provides an overview of the economic and epidemic modules, how they are coupled, which data we use to initialize them and what outputs they produce (Fig. 6a). A longer description that gives all details and justifies all the assumptions can be found in the Supplementary Information.

### Geography

The epidemic–economic model focuses on the New York–Newark–Jersey City, NY–NJ–PA metropolitan statistical area (federal information processing standard code C3562) (NY MSA). The NY MSA includes the highly urbanized area of New York City (that is, the five boroughs of Manhattan, Bronx, Queens, Brooklyn and Staten Island), but also some rural and industrial areas. The total population, as of 2019, is 19,216,182 individuals, making up about 6% of the US population. Total GDP, as of 2019, is around US$1.5 trillion, a bit more than 7% of the US GDP. A map of the NY MSA is shown in Supplementary Fig. 5. The epidemic module exclusively considers the NY MSA. This is because, while case importation from other parts of the United States and from abroad is an important factor in the early phase of a pandemic, it has little impact once the local incidence grows[26]. Our model is initialized in a period when local incidence was already relevant (Supplementary

Section 4.1), and so it is a valid approximation to only consider the NY MSA. The economic module models in detail the NY MSA area and also features a simplified model of the rest of the United States. The NY MSA economy is deeply integrated with that of the United States, and this integration must be taken into account throughout the pandemic to properly estimate economic impacts.

### Agents

The main agents of the epidemic–economic ABM are the 416,442 individuals of a synthetic population that is representative of the NY MSA (Supplementary Section 3.3.5). The population size is determined by the availability of mobility data (see below). The agents are heterogeneous in several of their socioeconomic characteristics (Fig. 6b), including age, income, employment status, occupation, possibility to WFH and the census tract where they live. Individuals are grouped into 153,547 households, whose composition is consistent with census microdata. We derive socioeconomic characteristics of synthetic individuals from tables provided by the American Community Survey (ACS) and the Bureau of Labor Statistics (BLS), trying to get as many joint distributions of variables as we can. For instance, to study heterogeneous outcomes across socioeconomic groups, it is important that synthetic individuals have the correct joint distribution of income, occupation and industry; this is important to replicate the empirical fact that managers working in the finance industry earn high income, while food preparation workers in the restaurants' industry earn low income. To achieve this objective, we combine United States-level BLS tables reporting incomes by industry–occupation pair with spatially detailed ACS tables giving incomes down to the census tract level. More details on the synthetic population building algorithm and validation tests can be found in Supplementary Section 3.3.

In the economic module, we also treat industries as agents, considering a single representative firm per industry. We use the two-digit North American Industry Classification System (NAICS) level of aggregation, giving 20 different industries. Industries are mainly dependent on one another through the input–output network of consumption of intermediate goods. Since no official data for the NY MSA exist, we downloaded national data from the Bureau of Economic Analysis (BEA) and then used a regionalization method known as Flegg Location Quotient[27] to obtain an input–output table that distinguishes between the NY MSA and the rest of the United States. The main idea behind this method is that a region that is more specialized in some good or service (such as information or finance in NY) can just rely on itself to source that good or service, but if it is less specialized (as for manufacturing in NY) it is more likely to import that good or service from the other(s) region. More details on our reconstruction of the input–output table can be found in Supplementary Section 3.4.

### Epidemic module

The epidemic module is a standard epidemiological model that runs on top of a contact network extracted from Global Positioning System (GPS) location data. Except for the integration with the economic module, the basic model is the same as the one described in ref. 16. In this model, individuals interact through a contact network composed of four layers: (1) the community layer captures occasional interactions between individuals, for instance occurring in consumption venues; (2) the workplace layer captures interactions between workers; (3) the household layer captures interactions between household members and (4) the school layer captures interactions between children attending the same school. To initialize contacts in the community and workplace layers to the pre-pandemic situation, we use privacy-preserving location intelligence data provided by Cuebiq, merging information about visits to points of Interest with a large database by Foursquare that characterizes points of Interest. We devised a privacy-preserving algorithm to match Cuebiq users to synthetic individuals, mainly based on the census tract where they live (Supplementary Section 3.3.6). Our approach to reconstructing contacts is probabilistic: because we cannot observe colocation of individuals reliably in the data, we use mobility data to estimate the probability that any pair of individuals are in contact on a given day and in a certain venue (Supplementary Section 1). The contact networks are initialized using data on pre-pandemic mobility, and modified over time due to exogenous interventions, feedback from the economic module and fear of infection (as explained in the sections below).

We use a stochastic, discrete-time infection transmission model coupled to the contact network (Fig. 6c) that extends the classical susceptible, latent, infected, removed model. A susceptible individual ($S$) may become infected upon contacting an infectious individual, moving to the latent compartment ($L$). Three states describe individuals who are potentially infectious, each of them with their corresponding transmission rate: pre-symptomatic ($P_s$), with transmission rate $\beta_s$; infectious symptomatic, with rate $\beta$; and infectious asymptomatic ($I_A$), with rate $r\beta$. Contacts between infectious and susceptible individuals depend on the contact network estimated for each day. Therefore, the probability that a susceptible node $i$ gets infected by an infectious node $j$ in infectious compartment type and place $p$ is:

$$P(S_i + I_j \rightarrow L_i + I_j) = 1 - e^{-\beta_{type} w_{i,j,p}(t)\Delta t} \qquad (1)$$

where $\Delta t = 1$ day and $w_{i,j,p}$ is a weight that modulates the effectiveness of contacts in a given setting in terms of spreading. We assume that all locations within the same layer (schools, workplaces, households and community) have the same weight, except for indoor/outdoor spaces in the community layer (Supplementary Section 4.1).

Once infected, the individual will enter the incubation compartment ($L$) for $\epsilon$ days, during which they will be infected but not infectious yet. A latent individual will become infectious $\gamma$ days before the end of the incubation period, to account for pre-symptomatic transmission. Whether an individual becomes symptomatic or not depends on the age-specific symptomatic probability, $p$. Lastly, the individual will be removed (R) from the infectious pool according to an exponential process with rate $\mu^{-1}$, where $\mu$ is the average length of the infectious period in days. Note that the removed compartment does not imply recovery, only that the individual is no longer able to infect. After $\delta$ days, removed individuals might transition to the death compartment according to the empirical age-dependent infection fatality ratio. A new death is reported $T_n$ days after the actual event to account for notification delays.

### Economic module

We introduce a dynamic macro-economic model that is specifically suited to study the economic effects of COVID-19, both at the macro-level of industries and at the micro-level of individuals and households. Figure 6d shows the causal relations between the variables of the economic module. It distinguishes between variables that are exogenous to the economic module (blue rectangles), such as the epidemic trajectory or the shelter-in-place policies that lead to supply shocks, and endogenous variables. These are further distinguished into industry-level endogenous variables (black rectangles) and individual-level endogenous variables (red diamonds). Examples of industry-level variables include employment, output, consumption and total demand. The only individual-level variable that we consider is employment status, although we do consider other agent attributes (such as age) that are fixed within our simulation period. At every time step $t$:

1. Industries decide the size of the workforce they need based on their past employment, past demand and the current levels of restrictions (we do not consider labour shortages due to illness or quarantine, as they would be difficult to model and preliminary simulations showed that they were a second-order effect). Conditional on the restrictions and on the previous employment status of individuals, industries decide which specific workers they hire or fire uniformly at random

2. Individuals, grouped into households, decide their consumption demand based both on fixed attributes such as age or income and on variable outcomes such as the situation of the epidemic, and on their employment status (workers who lose their job may cut back on spending). Aggregating over agents produces a total consumption demand for each industry

3. Total final demand for each industry is obtained by summing up consumption demand, orders of intermediate goods from other industries and other components of final demand (including government expenditures, investments, imports and exports)

4. Finally, industries produce goods and services. Industries aim to produce as much as demanded, but their production can be limited by labour shortages (for simplicity, in this paper we do not consider intermediate inputs shortages, which were not a first-order effect in the first few months of the pandemic[17]). In the case of shortages, demand is rationed on a pro-rata basis across intermediate and final consumers

Following the literature on dynamic input–output models that estimate the economic impact of natural disasters[24,25], we assume constant prices. We think that this is a good assumption at least in the short-term management of a pandemic. For instance, empirically, in the United States, prices decreased by 1.3% on an annual rate in Q2 2020, which was the smallest price change in the last three years (National Income and Products Account table 1.1.7, ref. 28, accessed 23 April 2023). Thus, for simplicity, we do not consider prices in our model.

### Coupling the epidemic and economic modules

The epidemic and economic modules are strongly coupled, in the sense that at time $t$, both modules take as input the output that the other module generated at $t - 1$. More specifically, the economic module takes the

number of deaths reported in the epidemic module, $D_{t-1}$, as an input to compute reduction in consumption demand, as explained in the 'Fear of infection' section below. At the same time, the epidemic module takes the employment status of each individual in the synthetic population as input from the economic module. This information is used in the epidemic module as previously employed individuals who get fired can no longer get infected in the workplace, while previously unemployed individuals who are hired can get infected. From a technical point of view, the epidemic module is written in C, while the economic module is written in Python. To implement coupling between the two models, we use a Python–C application programming interface (API)[29] that makes it possible to initialize a Python interpreter from within a C run.

## Timeline

A time step in the epidemic–economic model corresponds to one calendar day. Time effectively begins on 12 February 2020. On this date, the epidemic module starts running in calendar time (Supplementary Section 4.1) and producing epidemic outcomes on a daily scale. The economic module starts in a steady state that represents the economic situation at the beginning of 2020. All our simulations finish at the end of June 2020, giving a total of 140 time steps. In the empirical scenario, where we use our model to reproduce what happened during the first wave of COVID-19 in the NY MSA, we impose a number of exogenous interventions (see below) to both the epidemic and economic modules on 16 March 2020. We choose that date for simplicity, as several interventions were imposed at different times by the states of New York and New Jersey from 9 to 23 March, and we see an abrupt change in the social dynamics in our data around that date[16]. We remove the economic interventions on 15 May 2020, again as an approximation to the actual relaxation of protective measures that took place in the NY MSA during spring 2020. Other measures, such as the closure of schools and working from home, are kept in place until the end of the simulation. Similarly, for the modelling of counterfactuals, we start protective measures at various times, ranging from 17 February to 30 March, but still relax them on 15 May 2020.

## Exogenous interventions

During the COVID-19 pandemic, governments and local authorities imposed a number of NPIs with the goal of reducing the epidemic spreading. In this paper, we consider three types of interventions:

1. Closure of economic activities. We assume that a certain fraction $s_{k,t}$ of industry $k$ at time $t$ can be exogenously shut down. This implies that a fraction $s_{k,t}$ of the in-person workers of industry $k$ cannot work at $t$, reducing economic output of industry $k$. In the epidemic module, we assume the same reduction $s_{k,t}$ of contacts in the industry $k$. In the empirical scenario, we reproduce the closures that occurred in the NY MSA in spring 2020 (ref. 2), and we name this set of closures 'non-essential'. We consider alternative closing strategies when studying counterfactuals, such as the closure of customer-facing industries only

2. Imposition of WFH. All workers who can WFH must do so. We assume that this has no impact on the economic module—a worker who can WFH is as productive at home as in the workplace—but it reduces contacts and so infections in the workplace in the epidemic module

3. School closures. All contacts between children going to school are removed, effectively cutting off schools from disease transmission and thus reducing overall infections. We do not consider the impact of this policy on economic outcomes, due to the difficulty of calibrating the productivity loss related to childcare

## Fear of infection

Further to government interventions, a distinctive hallmark of the COVID-19 pandemic has been individuals' behaviour change: Due to

the fear of the disease, several individuals reduced their in-person consumption and contacts in the community and workplace as the epidemic situation worsened. We follow the large literature in behavioural epidemiology[30–35], and in particular the approach introduced in ref. 36, letting behaviour change follow the functional form

$$\Lambda_t(\phi, D_{t-1}) = 1 - \exp(-\phi D_{t-1}), \qquad (2)$$

where $D_{t-1}$ is the number of daily reported deaths in the NY MSA on day $t-1$, and $\phi$ is a sensitivity parameter that we name fear of infection. When $D_{t-1} = 0$, $\Lambda_t = 0$, so there is no behaviour change. In contrast, when $D_{t-1}$ grows large, there is substantial behaviour change, as $\Lambda_t \to 1$. Since $D_{t-1} \geq 0$, $\forall\, t$, behaviour change increases with fear of infection. The exponential functional form in equation (2) corresponds to a non-linearity in behavioural response that gives larger weight to an initial increase in $D_{t-1}$, with saturation afterwards[36], in line with the behavioural response to the first wave (for a comparison between our approach and other modelling efforts for fear of infection during the COVID-19 pandemic, see Supplementary Section 2.4). For parsimony, we assume that fear of infection is constant over time. This implies that the dynamics of the behaviour change are driven by the evolution of the death rate.

While behaviour change in both the epidemic and economic modules is based on the functional form in equation (2), there are slight differences in how behaviour change affects consumption and workplace and community contacts.

- Reduction in consumption (effect in the economic module). During the COVID-19 pandemic, individuals reduced 'risky' consumption of customer-facing services such as restaurants, cinemas, hairdressers and so on. However, they did not reduce consumption of financial and real estate services due to fear of infections. Most people kept paying rent, and even increased consumption of some manufacturing goods such as houseware. Therefore, we assume that fear of infection only decreased consumption in customer-facing industries, using the following functional form

$$\Lambda_{t,k}^{\mathrm{ECO}} = \Lambda(\phi^{\mathrm{ECO}}, D_{t-1})\, \tau_k, \qquad (3)$$

where $\tau_k$ is an indicator that takes value $\tau_k = 1$ if industry $k$ is customer facing, and $\tau_k = 0$ if it is not. Supplementary Section 3.1.3 lists which industries are customer facing, and explains our classification. The parameter $\phi^{\mathrm{ECO}}$ is a fear of infection parameter specific to the economic module, as we detail below

- Reduction in community contacts (effect in the epidemic module). As individuals reduce consumption of customer-facing services, they also decrease their contacts in the community, which has an effect on the epidemic module. However, the reduction in consumption is not identical to the reduction in community contacts. For instance, individuals may order takeaway meals from restaurants, thus reducing contacts but not consumption. We assume that reductions in consumption and community contacts are proportional, letting fear of infection in the epidemic module be given by

$$\phi^{\mathrm{EPI}} = \phi^{\mathrm{ECO}}/\tilde{\phi}, \qquad (4)$$

- with $\tilde{\phi}$ as a parameter giving the proportion between the two fear of infection parameters (for how we calibrate these parameters, see Supplementary Section 4). The reduction in community contacts is then given by

$$\Lambda_{t,k}^{\mathrm{EPI}} = \Lambda(\phi^{\mathrm{EPI}}, D_{t-1})\, \tau_k. \qquad (5)$$

- In practice, we implement this reduction in community contacts by multiplying the weights $w_{i,j,p}$ in equation (1) by $1 - \Lambda_{t,k}^{\mathrm{EPI}}$, which is the new level of contacts as modified by fear of infection

- Reduction in workplace contacts (effect in the epidemic module). If WFH is not imposed by the government or local authority, individuals may nevertheless decide to WFH if they can. We assume that this is uniform across industries, so that the reduction in workplace contacts among individuals who can WFH is given by

$$\Lambda_{t,k}^{\text{EPI,work}} = \Lambda\left(\phi^{\text{EPI}}, D_{t-1}\right). \tag{6}$$

- As above, we implement this reduction in workplace contacts by multiplying the weights by $1 - \Lambda_{t,k}^{\text{EPI,work}}$. We assume no reduction in productivity from working from home−a worker who can WFH is as productive at home as in the workplace−so this has no effect on the economic module

### Calibration and stochasticity
For both the epidemic and economic results, uncertainty comes from (1) stochasticity in the simulation runs, namely inherent stochasticity of transmission in the susceptible, latent, infected, removed model and inherent stochasticity in the hiring/firing process of the economic module; (2) uncertainty over parameter values, as obtained from the Approximate Bayesian Computation calibration algorithm that we use to calibrate the seven parameters that we cannot pin down independently (Supplementary Section 4). This means that, in line with a Bayesian approach, we run simulations sampling from all parameter values accepted by the Approximate Bayesian Computation.

### Reporting summary
Further information on research design is available in the Nature Portfolio Reporting Summary linked to this article.

## Data availability
All the data used in the economic module are publicly available and can be obtained from the American Community Survey, Bureau of Economic Analysis, Bureau of Labor Statistics and other sources, as described in Supplementary Information. The mobility data used in the epidemic module are available from Cuebiq, available upon request submitted to https://www.cuebiq.com/about/data-for-good/. The places data are obtainable from the Foursquare API (https://foursquare.com/products/places, accessed 16 February 2021).

## Code availability
The code that has been used to obtain all the quantitative results in the paper is publicly available on Zenodo (https://doi.org/10.5281/zenodo.7946867). Given that we cannot share the mobility data, as explained in the data availability statement above, we created synthetic random mobility networks. Thus, we cannot expect the code to quantitatively reproduce the results in the paper. Despite this, we provide a demo with a sanity check that shows that the model behaves as expected in obtaining aggregate unemployment and infections as a consequence of different closure policies.

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

## Acknowledgements

M.P., R.M.d.R.C. and A.P. acknowledge funding from the James S. McDonnell Foundation Postdoctoral Fellowship Award. F.L. and J.D.F. acknowledge funding from Baillie Gifford and the Institute for New Economic Thinking at the Oxford Martin School. A.A. acknowledges support through the grant RYC2021-033226-I funded by MCIN/AEI/10.13039/501100011033 and the European Union 'NextGenerationEU/PRTR'. Y.M was partially supported by the Government of Aragon, Spain and 'ERDF A way of making Europe' through grant E36-23R: Física Estadística y No Lineal (FENOL), and by Ministerio de Ciencia e Innovación, Agencia Española de Investigación (MCIN/AEI/10.13039/501100011033) grant no. PID2020-115800GB-I00. M.A., M.C. and A.V. acknowledge the support of the from the HHS/CDC 6U01IP001137 and HHS/CDC 5U01IP0001137. E.M. acknowledges acknowledges support by Ministerio de Ciencia e Innovación, Agencia Española de Investigación (MCIN/AEI/10.13039/501100011033) grant PID2019-106811GB-C32 and the National Science Foundation grant no. 2218748. All authors acknowledge the use of the computational resources of the Complex Systems & Networks (COSNET) Lab at the Institute for Biocomputation and Physics of Complex Systems (BIFI), funded by Banco Santander through grant Santander-UZ 2020/0274 and by the Government of Aragón (FONDO-COVID19-UZ-164255). The funders had no role in study design, data collection and analysis, decision to publish or preparation of the manuscript.

## Author contributions

M.P. proposed the collaboration. All authors designed the research. M.P. and A.A. performed the research, with contributions from R.M.d.R.C., A.P. and D.M.-C. M.P. and A.A. wrote the first draft. All authors contributed to the interpretation of the results, writing and editing of the paper.

## Competing interests

The authors declare no competing interests.

## Additional information

**Correspondence and requests for materials** should be addressed to Marco Pangallo.

# Reporting Summary

## Statistics

For all statistical analyses, confirm that the following items are present in the figure legend, table legend, main text, or Methods section.

| n/a | Confirmed | |
|---|---|---|
| ☐ | ☒ | The exact sample size (*n*) for each experimental group/condition, given as a discrete number and unit of measurement |
| ☒ | ☐ | A statement on whether measurements were taken from distinct samples or whether the same sample was measured repeatedly |
| ☐ | ☒ | The statistical test(s) used AND whether they are one- or two-sided<br>*Only common tests should be described solely by name; describe more complex techniques in the Methods section.* |
| ☒ | ☐ | A description of all covariates tested |
| ☒ | ☐ | A description of any assumptions or corrections, such as tests of normality and adjustment for multiple comparisons |
| ☐ | ☒ | A full description of the statistical parameters including central tendency (e.g. means) or other basic estimates (e.g. regression coefficient) AND variation (e.g. standard deviation) or associated estimates of uncertainty (e.g. confidence intervals) |
| ☐ | ☒ | For null hypothesis testing, the test statistic (e.g. *F*, *t*, *r*) with confidence intervals, effect sizes, degrees of freedom and *P* value noted<br>*Give P values as exact values whenever suitable.* |
| ☐ | ☒ | For Bayesian analysis, information on the choice of priors and Markov chain Monte Carlo settings |
| ☒ | ☐ | For hierarchical and complex designs, identification of the appropriate level for tests and full reporting of outcomes |
| ☒ | ☐ | Estimates of effect sizes (e.g. Cohen's *d*, Pearson's *r*), indicating how they were calculated |

*Our web collection on statistics for biologists contains articles on many of the points above.*

## Software and code

Policy information about availability of computer code

| Data collection | The data for both the economic and epidemic modules have been collected using standard software packages (Python and R). Census data have been obtained through the ACS API: https://cran.r-project.org/web/packages/acs/index.html.<br>We used Python 3.8.0, and the following versions of the Python packages: numpy==1.20.3; pandas==1.3.5; openpyxl==3.1.2.<br>We used R 4.3.1, and the following versions of the R packages: data.table_1.14.8, cowplot_1.1.1, tidycensus_1.4.4, latex2exp_0.9.6, gridExtra_2.3, zoo_1.8-12, stringr_1.4.0, ggrepel_0.9.1, ggplot2_3.3.5, reshape2_1.4.4, readxl_1.3.1, dplyr_1.0.7, sf_1.0-14 |
|---|---|
| Data analysis | The data for both the economic and epidemic modules have been analyzed using standard software packages (Python and R)<br>We used Python 3.8.0, and the following versions of the Python packages: numpy==1.20.3; pandas==1.3.5; openpyxl==3.1.2.<br>We used R 4.3.1, and the following versions of the R packages: data.table_1.14.8, cowplot_1.1.1, tidycensus_1.4.4, latex2exp_0.9.6, gridExtra_2.3, zoo_1.8-12, stringr_1.4.0, ggrepel_0.9.1, ggplot2_3.3.5, reshape2_1.4.4, readxl_1.3.1, dplyr_1.0.7, sf_1.0-14 |

For manuscripts utilizing custom algorithms or software that are central to the research but not yet described in published literature, software must be made available to editors and reviewers. We strongly encourage code deposition in a community repository (e.g. GitHub). See the Nature Portfolio guidelines for submitting code & software for further information.

# Data

Policy information about availability of data

All manuscripts must include a data availability statement. This statement should provide the following information, where applicable:
- Accession codes, unique identifiers, or web links for publicly available datasets
- A description of any restrictions on data availability
- For clinical datasets or third party data, please ensure that the statement adheres to our policy

All the data used in the economic module are publicly available and can be obtained as described in the Supplementary Information.
ACS API: https://cran.r-project.org/web/packages/acs/index.html.
BLS: https://www.bls.gov/oes/special.requests/oesm19in4.zip, https://www.bls.gov/cex/2019/msas/norteast.pdf, https://www.bls.gov/webapps/legacy/cpswktab3.htm, https://www.bls.gov/cex/2019/combined/age.pdf,https://www.bls.gov/cex/tables.htm, https://www.bls.gov/cex/pce-concordance-2017.xlsx, https://www.bls.gov/cew/data.htm
BEA: https://www.bea.gov/industry/input-output-accounts-data, https://apps.bea.gov/itable/iTable.cfm?ReqID=70&step=1&acrdn=5, https://apps.bea.gov/industry/xls/underlying-estimates/PCEBridge 1997-2019 SUM.xlsx

The mobility data used in the epidemic module are available from Cuebiq, available upon request submitted to https://www.cuebiq.com/about/data-for-good/. The Foursquare data are available from the public Foursquare API:https://foursquare.com/products/places ( Accessed 16-02-2021). https://developer.foursquare.com/docs/build-withfoursquare/categories/ (Accessed: 09-12-2020).

# Human research participants

Policy information about studies involving human research participants and Sex and Gender in Research.

| Reporting on sex and gender | We do not report on sex and gender of the participants. |
|---|---|
| Population characteristics | See below. |
| Recruitment | See below. |
| Ethics oversight | The privacy-enhanced mobility data was collected by the company Cuebiq using anonymized records of GPS locations from users that opted-in to share the data anonymously through a General Data Protection Regulation (GDPR) and California Consumer Privacy Act (CCPA) compliant framework. Additionally, we obtained IRB exemption to use the mobility data from the MIT IRB office. (COUHES protocol #1812635935 and its extension #E-2962) |

Note that full information on the approval of the study protocol must also be provided in the manuscript.

# Field-specific reporting

Please select the one below that is the best fit for your research. If you are not sure, read the appropriate sections before making your selection.

☐ Life sciences   ☒ Behavioural & social sciences   ☐ Ecological, evolutionary & environmental sciences

For a reference copy of the document with all sections, see nature.com/documents/nr-reporting-summary-flat.pdf

# Behavioural & social sciences study design

All studies must disclose on these points even when the disclosure is negative.

| Study description | In this study real-world data have been used to initialize the attributes of individual agents in the agent-based simulation. In particular, mobility data have been used to initialize the contacts in the community and workplace layers of the epidemic module. Tabular census data have been used to initialize the features of synthetic individuals and national accounting data have been used to initialize the economic input-output structure. |
|---|---|
| Research sample | The main agents of the epidemic-economic ABM are the 416,442 individuals of a synthetic population that is representative of the NY MSA. The agents are heterogeneous in several of their socio-economic characteristics (Figure \ref{fig:figureMaM}B), including age, income, employment status, occupation, possibility to work from home, and the census tract where they live. Individuals are grouped into 153,547 households, whose composition is consistent with census microdata. We derive socio-economic characteristics of synthetic individuals from tables provided by the American Community Survey (ACS) and the Bureau of Labor Statistics (BLS), trying to get as many joint distributions of variables as we can. More details on the synthetic population building algorithm and validation tests can be found in the Supplementary Information.

In the economic module, we also treat industries as agents, considering a single representative firm per industry. We use the 2-digit NAICS level of aggregation, giving 20 different industries. Industries are mainly dependent on one another through the input-output network of consumption of intermediate goods. Since no official data for the NY MSA exist, we downloaded national data from the |

| | |
|---|---|
| | Bureau of Economic Analysis (BEA) and then used a regionalization method to obtain an input-output table that distinguishes between the NY MSA and the Rest of the US. |
| Sampling strategy | The synthetic population size is determined by the availability of mobility data, collected from the geo-locations of 316070 anonymous opted-in devices collected by the company Cuebiq in the New York Metro Area. Post-stratification techniques has been used to correct for potential biases in the sample of users and to ensure population representation (see Supp. Materials 3.1.1). In particular, because the complete sample of users is slightly biased towards higher income individuals, we downsampled the original sample of users in the Cuebiq dataset to get a more representative distribution of the different quartiles of income. Specifically, the original sample contained 438,178 users and was biased toward high-income people. The penetration rate of high income people was 4.13\% while we only get 2.3\% of people in low-income areas. Thus we downsample the high-income groups to get a more balanced distribution of users by quantile groups. In particular we selected a random sample of 2.3\% of people in each of the income groups. This led us to a final set of 316,070 users. This does not correspond to the size of the synthetic population, as we need to include other agents that are not included in the Cuebiq data. For instance, we need to include children (for ethical and privacy reasons, we cannot access data for individuals less than 18 years old). We obtain the final population as described in Supplementary Sections S3.3.1 and S3.3.6.

There was no sampling involved for the industry agents - the entire US economy is represented in our model. |
| Data collection | The main source for the synthetic population is the American Community Survey (ACS). We used the R API that can be downloaded here: https://cran.r-project.org/web/packages/acs/index.html.

We then collected several labor and consumption data from the Bureau of Labor Statistics (BLS).
-income by occupation and industry (https://www.bls.gov/oes/special.requests/oesm19in4.zip)
-income by age (https://www.bls.gov/webapps/legacy/cpswktab3.htm, https://www.bls.gov/cex/2019/combined/age.pdf,)
-consumption across different categories (https://www.bls.gov/cex/2019/msas/norteast.pdf)
-consumption by age and income (https://www.bls.gov/cex/tables.htm)
-concordance between BLS and BEA consumption categories (https://www.bls.gov/cex/pce-concordance-2017.xlsx)
-Quarterly Census of Employment and Wages (https://www.bls.gov/cew/data.htm)

We finally collected data on input-output tables and the New York metro area regional economy from the Bureau of Economic Analysis (BEA):
-input-output data (https://www.bea.gov/industry/input-output-accounts-data).
-regional accounts (https://apps.bea.gov/itable/iTable.cfm?ReqID=70&step=1&acrdn=5)
-bridge between BEA consumption and production accounts (https://apps.bea.gov/industry/xls/underlying-estimates/PCEBridge 1997-2019 SUM.xlsx)

Data collection on mobility was done by the company Cuebiq, rfom the geo-locations of anonymous opted-in mobile phone devices .

We collected the Foursquare data from the public API: https://foursquare.com/products/places ( Accessed 16-02-2021). |
| Timing | Mobility data were collected from February 17, 2020 to June 30, 2020. The other data are from year 2019, to represent the pre-pandemic situation as accurately as possible. |
| Data exclusions | We only retain users that we could identify for the full period, i.e. that were observed in the last two months of the period (May-June 2020) at least once, as in Aleta et al., PNAS, 2022 (Ref [16] in the main text). This leads us to the sample containing 438,178 users mentioned above, which is then reduced to 316,070 users for representativeness reasons. |
| Non-participation | N/A: we did not solicit participation to this study |
| Randomization | N/A: this is not an experimental/empirical paper with an experimental and control group. |

# Reporting for specific materials, systems and methods

We require information from authors about some types of materials, experimental systems and methods used in many studies. Here, indicate whether each material, system or method listed is relevant to your study. If you are not sure if a list item applies to your research, read the appropriate section before selecting a response.

## Materials & experimental systems

| n/a | Involved in the study |
|---|---|
| ☒ | ☐ Antibodies |
| ☒ | ☐ Eukaryotic cell lines |
| ☒ | ☐ Palaeontology and archaeology |
| ☒ | ☐ Animals and other organisms |
| ☒ | ☐ Clinical data |
| ☒ | ☐ Dual use research of concern |

## Methods

| n/a | Involved in the study |
|---|---|
| ☒ | ☐ ChIP-seq |
| ☒ | ☐ Flow cytometry |
| ☒ | ☐ MRI-based neuroimaging |

