## [Peer Review File · Nature Human Behaviour]

Peer Review Information

Journal: Nature Human Behaviour

Manuscript Title: The unequal effects of the health-economy tradeoff during the COVID-19 pandemic

Corresponding author name(s): Marco Pangallo

Reviewer Comments & Decisions:

Decision Letter, initial version:
--

2nd March 2023

Dear Dr Pangallo,

Thank you once again for your manuscript, entitled "The unequal effects of the health-economy tradeoff during the COVID-19 pandemic", and for your patience during the peer review process.

Your Article has now been evaluated by 3 referees. You will see from their comments copied below that, although they find your work of potential interest, they have raised quite substantial concerns. In light of these comments, we cannot accept the manuscript for publication, but would be interested in considering a revised version if you are willing and able to fully address reviewer and editorial concerns.

We hope you will find the referees' comments useful as you decide how to proceed. If you wish to submit a substantially revised manuscript, please bear in mind that we will be reluctant to approach the referees again in the absence of major revisions. We are committed to providing a fair and constructive peer-review process. Do not hesitate to contact us if there are specific requests from the reviewers that you believe are technically impossible or unlikely to yield a meaningful outcome.

Editorially, we ask that in addition to addressing all reviewers' comments (focussing on Reviewer #2) you deposit your code to GitHub and provide a user friendly documentation for review and future users. Essentially, your README file should include a set of simple steps for us to reproduce sample results (with your synthetic network data). Please follow <https://www.nature.com/documents/GuidelinesCodePublication.pdf>.

Finally, your revised manuscript must comply fully with our editorial policies and formatting requirements. Failure to do so will result in your manuscript being returned to you, which will delay its consideration. To assist you in this process, I have attached a checklist that lists all of our requirements. If you have any questions about any of our policies or formatting, please don't hesitate

to contact me.

If you wish to submit a suitably revised manuscript, we would hope to receive it within 2 months. I would be grateful if you could contact us as soon as possible if you foresee difficulties with meeting this target resubmission date.

- Include a "Response to the editors and reviewers" document detailing, point-by-point, how you addressed each editor and referee comment. If no action was taken to address a point, you must provide a compelling argument. When formatting this document, please respond to each reviewer comment individually, including the full text of the reviewer comment verbatim followed by your response to the individual point. This response will be used by the editors to evaluate your revision and sent back to the reviewers along with the revised manuscript.
- Highlight all changes made to your manuscript or provide us with a version that tracks changes.

[REDACTED]

Thank you for the opportunity to review your work. Please do not hesitate to contact me if you have any questions or would like to discuss the required revisions further.

Sincerely,

Arunas Radzvilavicius, PhD
Editor, Nature Human Behaviour
Nature Research

Reviewer expertise:

Reviewer #1: mathematical epidemiology

Reviewer #2: economics, agent based models

Reviewer #3: epidemiology, computational modelling

REVIEWER COMMENTS:

Reviewer #1:

Remarks to the Author:

Dear Editors of NHB,

The manuscript by Pangallo et al is extremely important and has an important potential to provide useful information to PH and Economics policy makers.

Last but not least, scientifically it is very sound and well and analytically described in the supplementary materials (66 pages!).

Thus I recommend its publication, apart some minor revision.

Some minor points to be considered by authors are the following:

1. The "economic validation" subsection refers to a number of results illustrated in figures in the supplementary materials. Authors ought to consider to move in the main text some of them, because they refer to very important results
2. I did not understand why the "materials and methods" section is put after the bibliography. Is this a requirement asked by NHB? In my opinion the description of the model is as important as the description of results
3. References to previous works in modeling behavioral epidemiology of infectious diseases reduces to ref 20. On the contrary there is a vast literature in this field
4. Bibliography ends with ref 24 but further reference number are cited in the text. For example: reference 27 is cited in the right column of pag 11 just before and also a little bit later formula 2; reference 25 is cited in the first column of page 10; ref 26 is cited in the first row of the right column of page 10. They are in the supplementary materials, ok, but the material and methods is in the main text!

Kind Regards

An Anonymous Referee

Reviewer #2:

Remarks to the Author:

This paper presents a detailed model to study the impact of the COVID-19 pandemic. It focuses on the metropolitan statistical area New York-Newark-Jersey City. It includes a detailed epidemic model, incorporating a data-driven contact network capturing the individual interactions occurring in the household, in school, in the workplace and in the community. This epidemic model provides a very detailed account of the diffusion of the disease, through the relevant daily personal interactions. The pandemic model is augmented by an economic module, which allows to simulate the main macroeconomic variables such as unemployment, GDP, consumption.

Merging the pandemic and economic modules is very relevant. If they accounted only for the pandemic module, the best action would be to close everything in order to minimize deaths. If they accounted only for the economic module, the best action would be to keep everything open in order to minimize unemployment. Taking into account both modules allows to analyse the trade-off between curbing the pandemic and keeping the economy working. There is now a large literature (cited in the paper), triggered by the COVID-19 crisis, that tries to analyse the interactions between pandemic evolutions and economic outcomes, and aimed at evaluating the impact of the pandemics on economic performances. In this literature, a very relevant question is how to design policies capable of reducing deaths, but keeping the economy working.

I think that this paper does an excellent job in providing a model which can account for a very detailed analysis of the pandemics. The economic part is less detailed, but it does a good job in taking into

account the main variables of interests. For this reason, I believe that the paper has the potential to provide a significant and interesting advance in the understanding of the impact of pandemics on the economy.

The main results of the paper are the following:

I. It presents a model capable of reproducing the pandemic and economic dynamics to a fair approximation

II. It studies a set of counterfactuals, finding the following results:

a. High "fear of infection" has the same effect as imposed lockdowns (NPIs)

b. Closing only customer-facing industries is enough to curb the pandemic, keeping the economic damage to a lower level with respect to closing all non-essential industries (e.g., including manufacturing)

c. The timing of the intervention has a big impact on the spreading of the disease and on death count and a low impact on economic damage. So, better close earlier than later.

d. Finally, since the low-income workers are especially concentrated in customer-facing sectors and have tasks that usually cannot be performed from home, closures and fear of infection behaviour is affecting low income workers the most.

My comments to authors:

1) In the economic validation paragraph, the authors claim to be able to reproducing "some empirical properties on which [the model] has not been calibrated". These are: i. industry specific changes in employment and ii. low-income workers are more likely to be unemployed. It seems to me that these results are built in the calibration of the shocks. They are not "calibrated", but as the authors mention in the same paragraph, they are reproduced thanks to the estimation of the shocks, so they are somehow a direct consequence of the inputs given to the model. This is not wrong, but I wouldn't claim that the "model [...] predicts some empirical properties on which it has not been calibrated".

2) In the first paragraph of the discussion, the authors claim to be contributing to the debate on the efficacy of NPIs with respect to the behavioural response caused by the "fear of infection". However, I don't think this is the case. There are two key points to be addressed on this issue:

i. Is the "fear of infection" strong enough to curb the pandemic and avoid the negative externalities associated with individual behaviour neglecting the impact of their actions on the society? Essentially, if the fear of infection behaviour is strong enough, it leads people to stay at home even without any policy intervention and to behave as if they were internalizing the impact of their behaviour on the spread of the pandemic. Thus, the point to be discussed is whether this behavioural response is strong or weak. The paper does not have an answer to this point. It finds that a strong behavioural response has similar effects as NPIs, which is the premise of the debate. If it strong enough, then there is no need for policy intervention, but is it strong enough?

ii. The second important point to address is the economic implications of NPIs versus individual behavioural response. The point can be summarized as follows. Individual decisions maximize individual welfare. If the individual welfare does not internalize externalities, the individual maximization leads to poor aggregate outcomes. In this case it would be better to facilitate (or impose) coordination. In this particular case, if the fear of infection is too low, individual decisions would lead to a high spread of the disease. On the other hand, if the fear of infection is high, individual decisions would actually lead to a low spread of the disease. Now, is there any trade off? Is letting individuals decide how to behave providing any benefit (e.g., is there any gain in efficiency)? If it is not providing any benefit, then we should always adopt NPIs. I am not sure if there is any benefit in letting individuals decide their behaviour, and the paper does not provide an answer.

To summarize, the result on the impact of the fear of infection is a bit tautological and not very insightful. I find quite intuitive that fear of infections and NPIs have qualitatively similar results, and

the paper is not providing a deeper understanding of the issue. I would either downplay the result or provide a deeper analysis.

3) My take on the main policy implications of the paper are the following: i. regardless the “fear of infection parameter”, the timing of intervention is better earlier than later, ii. The policy intervention should impose the closure only of customer facing industries. I find these results interesting, but a bit underwhelming with respect to my expectations while I was reading the description of the model. The authors claim that the model is “mostly applicable to the short-term management of emerging/re-emerging infectious diseases”. This is true, but it does not provide any novel management method. Most NPIs after the first wave of COVID-19 did in fact close only customer-facing industries, and tried to act timely to curb the re-emerging of the disease. So, the strategies suggested by the paper were already adopted during the COVID pandemic (if not during the first wave, during the following ones). Clearly, the model gives a more formal base for such interventions, but yet, it does not provide any new management tool and I don’t think it exploits fully the capabilities of the model.

I suggest the authors to explore the following alternatives:

i. Given the detailed pandemic module, would it be possible to build a tool capable of driving the NPIs in a more detailed real-time way? Assume that you had real-time (or almost real-time) infections in the Metropolitan Area. Using your model and the data-driven contact network, you could try to forecast short-run spreading of the disease and design NPIs to try to contrast such spreading. The pandemic module can provide the forecasts on the spreading of the disease, while the economic model provides the cost of curbing the pandemic.

ii. Instead of analysing death, wouldn’t it be relevant to analyse the hospital occupancy rate? This would provide a much more direct way to evaluate different policies. The question would be: given available resources, i.e., number of ICUs, what policies minimize unemployment without overrunning the health system.

iii. In terms of different policies, the authors only analyse industry specific closures and timing of intervention in terms of date. If the aim is to provide a more general tool to respond to possible future pandemics, other important issues are to determine “when” to impose NPIs in terms of spreading of the disease (at what point of the diffusion dynamics should the NPIs be implemented?), and “for how long” (when should the NPIs be lifted?).

4) In Section S4.3 the authors describe the estimation technique used to select the value of a subset of parameters. There is technical error in the estimation method. In fact, the authors state that “We run each parameter combination with a different random seed”. Using a different seed for each parameter combination is increasing the estimation error. The correct estimation method uses the same random sequence for each parameter combination. The authors should use S simulations (as many as possible, taking into account computational resources) for each parameter combination using the same S seeds for each parameter combination.

5) I didn’t manage to run the code. If the paper is aimed at a wider audience, the code should be easier to run, or a more detailed readme should be provided.

Reviewer #3:

Remarks to the Author:

Review of: The unequal effects of the health-economy tradeoff during the COVID-19 Pandemic

This is a highly innovative and interdisciplinary paper addressing a theoretically deep and crucially important problem: The health-economy tradeoff in mitigating pandemic disease. The novel methods developed to address this—using coupled economic and epidemic modules—are quite general and will be useful in future crises. For this paper, the model is calibrated to the COVID-19 pandemic with high quality data from the greater New York metropolitan area. Not only does it reproduce a number of crucial economic and epidemic dynamics, but the calibrated model is then used to study several core scenarios and their impacts on disparate industries, socioeconomic groups, and geographical areas. It represents a very powerful application of high-resolution data-driven heterogeneous agent modeling to a complex—and highly politicized—problem. As such, it advances the science and disciplines the policy debate in this area.

Classical epidemic models ignore behavioral adaptation. There is now a recognition of its centrality and a powerful call for its inclusion, as in *Nature Human Behavior* (Bedson et al, 2022). That paper (on which I am a co-author) sounded the tocsin for better behavioral-epidemic modeling. This paper answers that call, advancing the field and fitting well (in my view) into NHB's emphasis on behavioral epidemiology. I have published widely on the need to include such cognitive drivers of behavior as contagious fear, which the authors represent in a convincing and transparent manner.

The data quality is high, and it strongly supports the conclusions. The results are highly significant, specifically regarding equity dimensions of non-pharmaceutical interventions such as workplace closures.

The references are thorough and literate on both the economic and epidemic dimensions of the problem. The writing is very clear throughout. What limitations I would note are noted by the authors themselves and certainly should not delay publication.

I have one suggestion to consider. The word "price" does not occur in the paper, nor do such related Economics terms as the "price elasticity of demand" or "income elasticity of demand." For example, the authors focus on reduction in consumer demand due to fear, which is a central connection. To an economist, however, such effects as reductions in demand are mediated by prices. For example, COVID supply-chain disruptions reduced supplies, raising prices, which is what reduced consumer demand. I think it might strengthen the paper, and broaden its audience, to say simply that they recognize the role of price dynamics, but that they are not modeled explicitly here, but are understood to operating 'under the hood,' as it were. Perhaps include a reference or two to standard economic modeling of price effects during COVID-19, just to demonstrate the literacy I know to be present for some of the authors who—full disclosure—are colleagues.

Overall, this is an extremely innovative paper on a crucially important topic and certainly should be published. Its focus on behavior makes it especially suitable for *Nature Human Behavior* and I sincerely hope to see it in print.

Author Rebuttal to Initial comments

1 Reviewer 1

Comment R1.1: *Dear Editors of NHB, The manuscript by Pangallo et al is extremely important and has an important potential to provide useful information to PH and Economics policy makers. Last but not least, scientifically it is very sound and well and analytically described in the supplementary materials (66 pages!). Thus I recommend its publication, apart some minor revision.*

Answer R1.1: We are truly honored for the positive evaluation of our paper.

Comment R1.2: *Some minor points to be considered by authors are the following: The “economic validation” subsection refers to a number of results illustrated in figures in the supplementary materials. Authors ought to consider to move in the main text some of them, because they refer to very important results.*

Answer R1.2: We thank the reviewer for considering the results in the supplementary information as very important. We now added some more details in the main text on the result on employment by occupation that is reported in the supplementary information. Moreover, the extra analysis on the predictive power of the model with respect to the supply shocks (in response to Comment R2.2) adds results to the economic validation subsection.

Comment R1.3: *I did not understand why the “materials and methods” section is put after the bibliography. Is this a requirement asked by NHB? In my opinion the description of the model is as important as the description of results*

Answer R1.3: We apologize for the confusion. We corrected the order of the sections following the NHB checklist. Now the Materials and Methods are before the bibliography, immediately after the Discussion.

Comment R1.4: *References to previous works in modeling behavioral epidemiology of infectious diseases reduces to ref 20. On the contrary there is a vast literature in this field*

Answer R1.4: We apologize for the omission. We have revised our analysis of the literature and added references 28-33, which we think fairly represent the large literature in behavioral epidemiology.

Comment R1.5: *Bibliography ends with ref 24 but further reference number are cited in the text. For example: reference 27 is cited in the right column of pag 11 just before and also a little bit later formula 2; reference 25 is cited in the first column of page 10; ref 26 is cited in the first row of the right column of page 10. They are in the supplementary materials, ok, but the material and methods is in the main text!*

Answer R1.5: We apologize for this issue. The problem was caused by the wrong order of the sections of the main text. All references in the main text are now included in the main bibliography.

2 Reviewer 2

Comment R2.1: *This paper presents a detailed model to study the impact of the COVID-19 pandemic. It focuses on the metropolitan statistical area New York-Newark-Jersey City. It*

includes a detailed epidemic model, incorporating a data-driven contact network capturing the individual interactions occurring in the household, in school, in the workplace and in the community. This epidemic model provides a very detailed account of the diffusion of the disease, through the relevant daily personal interactions. The pandemic model is augmented by an economic module, which allows to simulate the main macroeconomic variables such as unemployment, GDP, consumption. Merging the pandemic and economic modules is very relevant. If they accounted only for the pandemic module, the best action would be to close everything in order to minimize deaths. If they accounted only for the economic module, the best action would be to keep everything open in order to minimize unemployment. Taking into account both modules allows to analyse the trade-off between curbing the pandemic and keeping the economy working. There is now a large literature (cited in the paper), triggered by the COVID-19 crisis, that tries to analyse the interactions between pandemic evolutions and economic outcomes, and aimed at evaluating the impact of the pandemics on economic performances. In this literature, a very relevant question is how to design policies capable of reducing deaths, but keeping the economy working. I think that this paper does an excellent job in providing a model which can account for a very detailed analysis of the pandemics. The economic part is less detailed, but it does a good job in taking into account the main variables of interests. For this reason, I believe that the paper has the potential to provide a significant and interesting advance in the understanding of the impact of pandemics on the economy. The main results of the paper are the following: I. It presents a model capable of reproducing the pandemic and economic dynamics to a fair approximation II. It studies a set of counterfactuals, finding the following results: a. High “fear of infection” has the same effect as imposed lockdowns (NPIs) b. Closing only customer-facing industries is enough to curb the pandemic, keeping the economic damage to a lower level with respect to closing all non-essential industries (e.g., including manufacturing) c. The timing of the intervention has a big impact on the spreading of the disease and on death count and a low impact on economic damage. So, better close earlier than later. d. Finally, since the low-income workers are especially concentrated in customer-facing sectors and have tasks that usually cannot be performed from home, closures and fear of infection behaviour is affecting low income workers the most.

Answer R2.1: We thank the reviewer for the positive evaluation of our study.

Comment R2.2: *In the economic validation paragraph, the authors claim to be able to reproducing “some empirical properties on which [the model] has not been calibrated”. These are: i. industry specific changes in employment and ii. low-income workers are more likely to be unemployed. It seems to me that these results are built in the calibration of the shocks. They are not “calibrated”, but as the authors mention in the same paragraph, they are reproduced thanks to the estimation of the shocks, so they are somehow a direct consequence of the inputs given to the model. This is not wrong, but I wouldn’t claim that the “model [.] predicts some empirical properties on which it has not been calibrated”.*

Answer R2.2: We agree with the reviewer that the sentence was unclear, and have changed it accordingly: “Our model is also validated against empirical properties that were not directly targeted in the parameter calibration procedure” Moreover, we analyzed to which extent the model was able to improve over the shocks in predicting industry-specific outcomes. To this end, we compared the predictions of the model to the predictions that would have been obtained by the supply shocks only, computing the fraction of workers in a given industry that cannot work from home and are not essential (see Ref. [2] in the main paper). We found that the Pearson correlation coefficient computed across industries between model estimates and empirical data is 0.82, higher than

the 0.69 correlation obtained when using shock predictions. This discrepancy is largely driven by certain industries, such as transportation, which despite being predominantly essential and thus experiencing minimal supply shock, experience employment reductions due to declines in final and intermediate demand. These dynamics are an integral part of the transmission mechanism of the economic module. We have incorporated this finding into the validation section and documented it with a new figure in the supplementary information. (Supplementary Figure S14, which is included below for reviewer’s convenience).

Figure 1: **Shocks and model vs. empirical data.** We compare the ratio of employment in April 2020 to the pre-pandemic situation between two predictions and empirical data, across the largest industries. Blue dots represent the predictions of the first-order supply shocks only, obtained by computing the workers that cannot work from home and are not essential according to the last column of Table S3. Red dots indicate the predictions by the model. The two predictions for the same industry are linked to ease comparison. The identity line is plotted for reference.

Comment R2.3: *In the first paragraph of the discussion, the authors claim to be contributing to the debate on the efficacy of NPIs with respect to the behavioural response caused by the “fear of infection”. However, I don’t think this is the case. There are two key points to be addressed on this issue: i. Is the “fear of infection” strong enough to curb the pandemic and avoid the negative externalities associated with individual behaviour neglecting the impact of their actions on the society? Essentially, if the fear of infection behaviour is strong enough, it leads people to stay at home even without any policy intervention and to behave as if they were internalizing the impact of their behaviour on the spread of the pandemic. Thus, the point to be discussed is whether this behavioural response is strong or weak. The paper does not have an answer to*

this point. It finds that a strong behavioural response has similar effects as NPIs, which is the premise of the debate. If it strong enough, then there is no need for policy intervention, but is it strong enough?

Answer R2.3: We agree with the reviewer that it is not possible to empirically estimate the fear of infection as a stand-alone parameter in the presence of policy interventions. Indeed, an identification issue arises as the fear parameter we calibrated conflates the impacts of infection fear and policy interventions, likely leading to its underestimation. Thus, we avoided attributing a causal interpretation to this fear estimation, choosing instead to examine scenarios with significantly higher or lower fear than the calibrated value. In high fear scenarios (more behavioral changes), we observed a greater reduction in deaths and a steeper rise in unemployment compared to closure scenarios, suggesting that intense infection fear could potentially suppress the pandemic more effectively than policy interventions, albeit at a higher economic cost. We have clarified this in the design of experiments section.

Comment R2.4: *The second important point to address is the economic implications of NPIs versus individual behavioural response. The point can be summarized as follows. Individual decisions maximize individual welfare. If the individual welfare does not internalize externalities, the individual maximization leads to poor aggregate outcomes. In this case it would be better to facilitate (or impose) coordination. In this particular case, if the fear of infection is too low, individual decisions would lead to a high spread of the disease. On the other hand, if the fear of infection is high, individual decisions would actually lead to a low spread of the disease. Now, is there any trade off? Is letting individuals decide how to behave providing any benefit (e.g., is there any gain in efficiency)? If it is not providing any benefit, then we should always adopt NPIs. I am not sure if there is any benefit in letting individuals decide their behaviour, and the paper does not provide an answer. To summarize, the result on the impact of the fear of infection is a bit tautological and not very insightful. I find quite intuitive that fear of infections and NPIs have qualitatively similar results, and the paper is not providing a deeper understanding of the issue. I would either downplay the result or provide a deeper analysis.*

Answer R2.4: We agree with the reviewer on the different interpretation of behavioral response and NPIs – behavioral response internalizes the pandemic risk, while NPIs impose coordination when individual behavioral response is not enough. However, we do not think that it is obvious that behavioral response and NPIs lead to the same trends and trade-offs in infection and unemployment rates.

First, the mechanisms by which behavioral response and NPIs operate are quite different. The effect of behavioral response is the outcome of a self-organizing process that becomes substantial when the number of notified deaths increases sufficiently, which can be considerably delayed compared to infections, and then leads to reduced consumption demand of goods and services provided by customer-facing industries. In contrast, NPIs can be imposed at any arbitrary point in time. Moreover, in our model NPIs operate by reducing the labor supply across potentially any industry, not just customer-facing ones. In economic terms, in our model behavioral response is a demand shock, while NPIs are a supply shock. It is not obvious that they should have the same aggregate effects.

Here, following the type of adaptive behavior that has been studied in most of the epidemiological literature, we show that, under these assumptions, behavioral response leads to the same outcomes of NPIs. Again, to us this was not obvious from the start, given the complexity of the model. For instance, it could have been that under the behavioral response rule that we considered, infections stopped suddenly and people would be able to resume consumption immediately, leading to the best epidemic and economic outcomes. Or, it could have been that under strong behavior change infections did not reduce so much, while consumption would decrease so much that the economy would collapse, leading to the worst epidemic and economic outcomes. None of these situations play out

in our validated, data-driven, epidemic-economic model, but it could play out in a more qualitative model that has not been appropriately validated and based on granular data.

We thank the reviewer for this comment, which allowed us to clarify these key points both in the results and the discussion sections.

Furthermore, by following the suggestion of the reviewer, we now provide a deeper analysis on the fear of infection. Specifically, we added a new composite figure in the main text and a new subsection of results. We started from the premise that, to be fair to the laissez-faire approach, we should let individuals internalize their own risk (ignoring the risk they pose to others if they become infected). To this end, we considered a theoretical scenario with age-specific fear of infection. In this scenario, households are grouped in three age classes, and each class has a level of fear that is proportional to the death risk in that class, in a way that aggregate fear is the same as in the scenario in which fear is uniform across age classes. As we describe in the new subsection, this scenario does not show a departure from the results obtained previously with the homogeneous fear of infection, strengthening the similarity between the effects of behavioral response and NPIs on public health and epidemic outcomes.

In general, although it was possible to anticipate some qualitative results regarding the fear of infection, we believe that a quantitative evaluation was still lacking. We believe that our modeling analysis, based on a model coupling detailed economic and demographic information with data-driven contact networks provides relevant insights on one of the crucial debates that was spurred by the COVID-19 pandemic.

Comment R2.5: *My take on the main policy implications of the paper are the following: i. regardless the “fear of infection parameter”, the timing of intervention is better earlier than later, ii. The policy intervention should impose the closure only of customer facing industries. I find these results interesting, but a bit underwhelming with respect to my expectations while I was reading the description of the model. The authors claim that the model is “mostly applicable to the short-term management of emerging/re-emerging infectious diseases”. This is true, but it does not provide any novel management method. Most NPIs after the first wave of COVID-19 did in fact close only customer-facing industries, and tried to act timely to curb the re-emerging of the disease. So, the strategies suggested by the paper were already adopted during the COVID pandemic (if not during the first wave, during the following ones). Clearly, the model gives a more formal base for such interventions, but yet, it does not provide any new management tool and I don’t think it exploits fully the capabilities of the model.*

Answer R2.5: On one hand, we believe that giving a more formal and quantitative basis to the interventions that were adopted in later waves is by itself a worthwhile result. Before our paper, there were no theoretical models grounded on detailed real-world data that could address the joint epidemic-economic effects of different policies across socio-economic groups, and we think that quantitatively testing these policies is an important contribution. For instance, as the referee notes, “most” NPIs did restrict customer-facing industries, but some governments decided to support customer-facing industries during summer 2020 (see e.g. the eat-out-to-help-out scheme in the UK: <https://doi.org/10.1093/ej/ueab074>). This shows the need for continued publication of rigorous scientific evaluation of these policies.

On the other hand we also found not so intuitive results such as the fact that implementing NPIs late leads to almost no decrease in unemployment as providing relevant information for epidemic management. Another relevant finding for pandemic management is indeed the fact that while NPIs and spontaneous behavioral adaptation have similar effects on public health and economic outcomes, only NPIs can be implemented as soon as needed for highest effectiveness.

We now discuss these points in the manuscript.

Comment R2.6: *I suggest the authors to explore the following alternatives: i. Given the detailed*

pandemic module, would it be possible to build a tool capable of driving the NPIs in a more detailed real-time way? Assume that you had real-time (or almost real-time) infections in the Metropolitan Area. Using your model and the data-driven contact network, you could try to forecast short-run spreading of the disease and design NPIs to try to contrast such spreading. The pandemic module can provide the forecasts on the spreading of the disease, while the economic model provides the cost of curbing the pandemic. iii. In terms of different policies, the authors only analyse industry specific closures and timing of intervention in terms of date. If the aim is to provide a more general tool to respond to possible future pandemics, other important issues are to determine “when” to impose NPIs in terms of spreading of the disease (at what point of the diffusion dynamics should the NPIs be implemented?), and “for how long” (when should the NPIs be lifted?).

Answer R2.6: We thank the reviewer for suggesting us to explore optimized policies and strategies for pandemic management. While the developed model lends itself to this job, designing realistic and feasible policies depends on assessing their practical feasibility and a case-by-case examination, based on resources, logistics as well as the objective functions to optimize, which is beyond this paper’s scope. Following the suggestion of the reviewer however we have now added a discussion of these points in both the results and discussion sections.

Comment R2.7: *ii. Instead of analysing death, wouldn’t it be relevant to analyse the hospital occupancy rate? This would provide a much more direct way to evaluate different policies. The question would be: given available resources, i.e., number of ICUs, what policies minimize unemployment without overrunning the health system.*

Answer R2.7: We agree with the reviewer that analyzing ICU occupancy could provide useful insights. However, a discussion of ICU occupancy requires several assumptions on the management of patients, and the ramping up of hospital capacity, that have been rapidly changing during the early months of the pandemic and then during the course of the following years. While the model can be applied to discuss specific situations, it would be difficult to extrapolate general results. We believe that such scenario analysis is worthwhile in the context of specific situation and health care objectives, but not in the scope of a paper presenting a general methodological approach.

Comment R2.8: *In Section S4.3 the authors describe the estimation technique used to select the value of a subset of parameters. There is technical error in the estimation method. In fact, the authors state that “We run each parameter combination with a different random seed”. Using a different seed for each parameter combination is increasing the estimation error. The correct estimation method uses the same random sequence for each parameter combination. The authors should use S simulations (as many as possible, taking into account computational resources) for each parameter combination using the same S seeds for each parameter combination.*

Answer R2.8: We re-ran the calibration procedure following the suggestion by the reviewer. Instead of running 100,000 parameter combinations with 1 random seed, we now run 10,000 parameter combinations with 10 random seeds each. We also re-ran all the analyses and replaced all the figures in the main text and in the supplementary information. Although we observe very small differences with respect to the originally submitted manuscript, the obtained conclusions are unchanged.

Comment R2.9: *I didn’t manage to run the code. If the paper is aimed at a wider audience, the code should be easier to run, or a more detailed readme should be provided.*

Answer R2.9: We apologize for the lack of detail. We have now provided a Docker image of the repository that should be easier to run. Code documentation has been improved as well.

3 Reviewer 3

Comment R3.1: *This is a highly innovative and interdisciplinary paper addressing a theoretically deep and crucially important problem: The health-economy tradeoff in mitigating pandemic disease. The novel methods developed to address this—using coupled economic and epidemic modules—are quite general and will be useful in future crises. For this paper, the model is calibrated to the COVID-19 pandemic with high quality data from the greater New York metropolitan area. Not only does it reproduce a number of crucial economic and epidemic dynamics, but the calibrated model is then used to study several core scenarios and their impacts on disparate industries, socioeconomic groups, and geographical areas. It represents a very powerful application of high-resolution data-driven heterogeneous agent modeling to a complex—and highly politicized—problem. As such, it advances the science and disciplines the policy debate in this area. Classical epidemic models ignore behavioral adaptation. There is now a recognition of its centrality and a powerful call for its inclusion, as in Nature Human Behavior (Bedson et al, 2022). That paper (on which I am a co-author) sounded the tocsin for better behavioral-epidemic modeling. This paper answers that call, advancing the field and fitting well (in my view) into NHB’s emphasis on behavioral epidemiology. I have published widely on the need to include such cognitive drivers of behavior as contagious fear, which the authors represent in a convincing and transparent manner. The data quality is high, and it strongly supports the conclusions. The results are highly significant, specifically regarding equity dimensions of non-pharmaceutical interventions such as workplace closures. The references are thorough and literate on both the economic and epidemic dimensions of the problem. The writing is very clear throughout. What limitations I would note are noted by the authors themselves and certainly should not delay publication.*

Answer R3.1: We thank the reviewer for very positive evaluation of our manuscript and for bringing the Bedson et al. paper to our attention. We included it in our reference list, and mention it at the end of the discussion where we summarize which literature this paper can contribute to advance.

Comment R3.2: *I have one suggestion to consider. The word “price” does not occur in the paper, nor do such related Economics terms as the “price elasticity of demand” or “income elasticity of demand.” For example, the authors focus on reduction in consumer demand due to fear, which is a central connection. To an economist, however, such effects as reductions in demand are mediated by prices. For example, COVID supply-chain disruptions reduced supplies, raising prices, which is what reduced consumer demand. I think it might strengthen the paper, and broaden its audience, to say simply that they recognize the role of price dynamics, but that they are not modeled explicitly here, but are understood to operating ‘under the hood,’ as it were. Perhaps include a reference or two to standard economic modeling of price effects during COVID-19, just to demonstrate the literacy I know to be present for some of the authors who—full disclosure—are colleagues.*

Answer R3.2: The reviewer brings up an important point, that a subset of the authors acknowledge in another

publication (ref. 17). It is definitely true that, in many cases of economic response to natural or human-made disasters that reduce supply, increases in prices mediate reductions in demand. This is what we saw, for instance, during the Russian invasion of Ukraine, when the price of Liquefied Natural Gas increased a lot, leading consumers and businesses in Europe to save on gas consumption and thus reduce demand, matching the reduced supply. However, we do not think that the price channel had any major role in the first wave of the COVID-19 pandemic, except for a few items that are unlikely to have macroeconomic impact (masks, sanitizers, etc). This is corroborated by macroeconomic statistics. For instance, checking the annualized percentage change in prices between Q2-2020 and Q1-2020 in NIPA table 1.1.7 (<https://apps.bea.gov/iTable/?reqid=19&step=3&isuri=1&1921=survey&1903=11>, accessed Sunday, April 23, 2023), we see that at the aggregate level it was the smallest change in the last three years.

More in general, while general equilibrium models used to estimate the economic impact of the COVID-19 pandemic (such as ref. 11 in the main text) mediate changes in demand through prices, dynamic input-output models from the natural disasters literature (such as refs 24-25 in the main text) do not. These papers focus on a very short period, during which prices have little time to change and rationing is more likely to play a first-order effect.

We now clarify why we do not consider price changes with a paragraph in Materials and Methods.

Comment R3.3: *Overall, this is an extremely innovative paper on a crucially important topic and certainly should be published. Its focus on behavior makes it especially suitable for Nature Human Behavior and I sincerely hope to see it in print.*

Answer R3.3: We thank again the reviewer for the positive evaluation of our work.

Decision Letter, first revision:

22nd May 2023

Dear Dr Pangallo,

Thank you for submitting your revised manuscript, "The unequal effects of the health-economy tradeoff during the COVID-19 pandemic". After careful consideration and discussion with my colleagues, I am sorry to have to tell you that we do not feel that the referees' comments have been sufficiently addressed to justify sending this revision back for peer review.

This unusual course of action is taken occasionally to avoid unproductive rounds of review that result in reviewer fatigue and damage the chances of the manuscript obtaining a fair and objective evaluation. Such situations are not in an author's best interest so we try to avoid them when it seems prudent to do so.

In order to consider this manuscript further we would request that you please do your best to fully address all of the comments of the reviewers. In particular, our view is that Reviewer points 2.6 and 2.7 need to be addressed by implementing the analyses suggested. I understand that designing specific policies requires case-by-case assessment, and that these steps are context dependent. However, we believe that usefulness of the model depends on you being able to demonstrate that it can be applied to making policy decisions in specific contexts, regarding the timing of interventions as well as taking into account ICU occupancy, even if this does not generalize.

Should you be able to adequately respond to these and the reviewers' other concerns, we would be happy to look at a revised manuscript.

We shall hope to receive your revised version as soon as possible. If you anticipate a delay of more than four weeks, however, please let us know. We will be happy to consider your revision so long as nothing similar has been accepted for publication at Nature Human Behaviour or published elsewhere. Should your manuscript be substantially delayed without notifying us in advance and your article is eventually published, the received date may be that of the revised, not the original, version.

Nature Human Behaviour is committed to improving transparency in authorship. As part of our efforts in this direction, we are now requesting that all authors identified as 'corresponding author' on published papers create and link their Open Researcher and Contributor Identifier (ORCID) with their account on the Manuscript Tracking System (MTS), prior to acceptance. ORCID helps the scientific community achieve unambiguous attribution of all scholarly contributions. You can create and link your ORCID from the home page of the MTS by clicking on 'Modify my Springer Nature account'. For more information please visit www.springernature.com/orcid.

If you are not interested in submitting a suitably revised manuscript in the future please let me know immediately so we can close your file. If you have any questions, please contact me.

Please use the link below to submit a suitably revised manuscript and updated response to referees when they are ready.

[REDACTED]

Sincerely,

Arunas Radzvilavicius, PhD
Senior Editor, Nature Human Behaviour
Nature Research

Author Rebuttal, first revision:

1 Reviewer 1

Comment R1.1: *Dear Editors of NHB, The manuscript by Pangallo et al is extremely important and has an important potential to provide useful information to PH and Economics policy makers. Last but not least, scientifically it is very sound and well and analytically described in the supplementary materials (66 pages!). Thus I recommend its publication, apart some minor revision.*

Answer R1.1: We are truly honored for the positive evaluation of our paper.

Comment R1.2: *Some minor points to be considered by authors are the following: The “economic validation” subsection refers to a number of results illustrated in figures in the supplementary materials. Authors ought to consider to move in the main text some of them, because they refer to very important results.*

Answer R1.2: We thank the reviewer for considering the results in the supplementary information as very important. We now added some more details in the main text on the result on employment by occupation that is reported in the supplementary information. Moreover, the extra analysis on the predictive power of the model with respect to the supply shocks (in response to Comment R2.2) adds results to the economic validation subsection.

Comment R1.3: *I did not understand why the “materials and methods” section is put after the bibliography. Is this a requirement asked by NHB? In my opinion the description of the model is as important as the description of results*

Answer R1.3: We apologize for the confusion. We corrected the order of the sections following the NHB checklist. Now the Materials and Methods are before the bibliography, immediately after the Discussion.

Comment R1.4: *References to previous works in modeling behavioral epidemiology of infectious diseases reduces to ref 20. On the contrary there is a vast literature in this field*

Answer R1.4: We apologize for the omission. We have revised our analysis of the literature and added references 28-33, which we think fairly represent the large literature in behavioral epidemiology.

Comment R1.5: *Bibliography ends with ref 24 but further reference number are cited in the text. For example: reference 27 is cited in the right column of pag 11 just before and also a little bit later formula 2; reference 25 is cited in the first column of page 10; ref 26 is cited in the first row of the right column of page 10. They are in the supplementary materials, ok, but the material and methods is in the main text!*

Answer R1.5: We apologize for this issue. The problem was caused by the wrong order of the sections of the main text. All references in the main text are now included in the main bibliography.

2 Reviewer 2

Comment R2.1: *This paper presents a detailed model to study the impact of the COVID-19 pandemic. It focuses on the metropolitan statistical area New York-Newark-Jersey City. It includes a detailed epidemic model, incorporating a data-driven contact network capturing the individual interactions occurring in the household, in school, in the workplace and in the community. This epidemic model provides a very detailed account of the diffusion of the disease, through the relevant daily personal interactions. The pandemic model is augmented by an economic module, which allows to simulate the main macroeconomic variables such as unemployment, GDP, consumption. Merging the pandemic and economic modules is very relevant. If they accounted only for the pandemic module, the best action would be to close everything in order to minimize deaths. If they accounted only for the economic module, the best action would be to keep everything open in order to minimize unemployment. Taking into account both modules allows to analyse the trade-off between curbing the pandemic and*

keeping the economy working. There is now a large literature (cited in the paper), triggered by the COVID-19 crisis, that tries to analyse the interactions between pandemic evolutions and economic outcomes, and aimed at evaluating the impact of the pandemics on economic performances. In this literature, a very relevant question is how to design policies capable of reducing deaths, but keeping the economy working. I think that this paper does an excellent job in providing a model which can account for a very detailed analysis of the pandemics. The economic part is less detailed, but it does a good job in taking into account the main variables of interests. For this reason, I believe that the paper has the potential to provide a significant and interesting advance in the understanding of the impact of pandemics on the economy. The main results of the paper are the following: I. It presents a model capable of reproducing the pandemic and economic dynamics to a fair approximation II. It studies a set of counterfactuals, finding the following results: a. High “fear of infection” has the same effect as imposed lockdowns (NPIs) b. Closing only customer-facing industries is enough to curb the pandemic, keeping the economic damage to a lower level with respect to closing all non-essential industries (e.g., including manufacturing) c. The timing of the intervention has a big impact on the spreading of the disease and on death count and a low impact on economic damage. So, better close earlier than later. d. Finally, since the low-income workers are especially concentrated in customer-facing sectors and have tasks that usually cannot be performed from home, closures and fear of infection behaviour is affecting low income workers the most.

Answer R2.1: We thank the reviewer for the positive evaluation of our study.

Comment R2.2: *In the economic validation paragraph, the authors claim to be able to reproducing “some empirical properties on which [the model] has not been calibrated”. These are: i. industry specific changes in employment and ii. low-income workers are more likely to be unemployed. It seems to me that these results are built in the calibration of the shocks. They are not “calibrated”, but as the authors mention in the same paragraph, they are reproduced thanks to the estimation of the shocks, so they are somehow a direct consequence of the inputs given to the model. This is not wrong, but I wouldn’t claim that the “model [.] predicts some empirical properties on which it has not been calibrated”.*

Answer R2.2: We agree with the reviewer that the sentence was unclear, and have changed it accordingly: “Our model is also validated against empirical properties that were not directly targeted in the parameter calibration procedure” Moreover, we analyzed to which extent the model was able to improve over the shocks in predicting industry-specific outcomes. To this end, we compared the predictions of the model to the predictions that would have been obtained by the supply shocks only, computing the fraction of workers in a given industry that cannot work from home and are not essential (see Ref. [2] in the main paper). We found that the Pearson correlation coefficient computed across industries between model estimates and empirical data is 0.82, higher than the 0.69 correlation obtained when using shock predictions. This discrepancy is largely driven by certain industries, such as transportation, which despite being predominantly essential and thus experiencing minimal supply shock, experience employment reductions due to declines in final and intermediate demand. These dynamics are an integral part of the transmission mechanism of the economic module. We have incorporated this finding into the validation section and documented it with a new figure in the supplementary information. (Supplementary Figure S14, which is included below for reviewer’s convenience).

Figure 1: **Shocks and model vs. empirical data.** We compare the ratio of employment in April 2020 to the pre-pandemic situation between two predictions and empirical data, across the largest industries. Blue dots represent the predictions of the first-order supply shocks only, obtained by computing the workers that cannot work from home and are not essential according to the last column of Table S3. Red dots indicate the predictions by the model. The two predictions for the same industry are linked to ease comparison. The identity line is plotted for reference.

Comment R2.3: *In the first paragraph of the discussion, the authors claim to be contributing to the debate on the efficacy of NPIs with respect to the behavioural response caused by the “fear of infection”. However, I don’t think this is the case. There are two key points to be addressed on this issue: i. Is the “fear of infection” strong enough to curb the pandemic and avoid the negative externalities associated with individual behaviour neglecting the impact of their actions on the society? Essentially, if the fear of infection behaviour is strong enough, it leads people to stay at home even without any policy intervention and to behave as if they were internalizing the impact of their behaviour on the spread of the pandemic. Thus, the point to be discussed is whether this behavioural response is strong or weak. The paper does not have an answer to this point. It finds that a strong behavioural response has similar effects as NPIs, which is the premise of the debate. If it strong enough, then there is no need for policy intervention, but is it strong enough?*

Answer R2.3: We agree with the reviewer that it is not possible to empirically estimate the fear of infection as a stand-alone parameter in the presence of policy interventions. Indeed, an identification issue arises as the fear parameter we calibrated conflates the impacts of infection fear and policy interventions, likely leading to its underestimation. Thus, we avoided attributing a causal interpretation to this fear estimation, choosing instead to examine scenarios with significantly higher or lower fear than the calibrated value. In high fear scenarios (more behavioral changes), we observed a greater reduction in deaths and a steeper rise in unemployment compared to

closure scenarios, suggesting that intense infection fear could potentially suppress the pandemic more effectively than policy interventions, albeit at a higher economic cost. We have clarified this in the design of experiments section.

Comment R2.4: *The second important point to address is the economic implications of NPIs versus individual behavioural response. The point can be summarized as follows. Individual decisions maximize individual welfare. If the individual welfare does not internalize externalities, the individual maximization leads to poor aggregate outcomes. In this case it would be better to facilitate (or impose) coordination. In this particular case, if the fear of infection is too low, individual decisions would lead to a high spread of the disease. On the other hand, if the fear of infection is high, individual decisions would actually lead to a low spread of the disease. Now, is there any trade off? Is letting individuals decide how to behave providing any benefit (e.g., is there any gain in efficiency)? If it is not providing any benefit, then we should always adopt NPIs. I am not sure if there is any benefit in letting individuals decide their behaviour, and the paper does not provide an answer. To summarize, the result on the impact of the fear of infection is a bit tautological and not very insightful. I find quite intuitive that fear of infections and NPIs have qualitatively similar results, and the paper is not providing a deeper understanding of the issue. I would either downplay the result or provide a deeper analysis.*

Answer R2.4: We agree with the reviewer on the different interpretation of behavioral response and NPIs – behavioral response internalizes the pandemic risk, while NPIs impose coordination when individual behavioral response is not enough. However, we do not think that it is obvious that behavioral response and NPIs lead to the same trends and trade-offs in infection and unemployment rates.

First, the mechanisms by which behavioral response and NPIs operate are quite different. The effect of behavioral response is the outcome of a self-organizing process that becomes substantial when the number of notified deaths increases sufficiently, which can be considerably delayed compared to infections, and then leads to reduced consumption demand of goods and services provided by customer-facing industries. In contrast, NPIs can be imposed at any arbitrary point in time. Moreover, in our model NPIs operate by reducing the labor supply across potentially any industry, not just customer-facing ones. In economic terms, in our model behavioral response is a demand shock, while NPIs are a supply shock. It is not obvious that they should have the same aggregate effects.

Here, following the type of adaptive behavior that has been studied in most of the epidemiological literature, we show that, under these assumptions, behavioral response leads to the same outcomes of NPIs. Again, to us this was not obvious from the start, given the complexity of the model. For instance, it could have been that under the behavioral response rule that we considered, infections stopped suddenly and people would be able to resume consumption immediately, leading to the best epidemic and economic outcomes. Or, it could have been that under strong behavior change infections did not reduce so much, while consumption would decrease so much that the economy would collapse, leading to the worst epidemic and economic outcomes. None of these situations play out in our validated, data-driven, epidemic-economic model, but it could play out in a more qualitative model that has not been appropriately validated and based on granular data.

We thank the reviewer for this comment, which allowed us to clarify these key points both in the results and the discussion sections.

Furthermore, by following the suggestion of the reviewer, we now provide a deeper analysis on the fear of infection. Specifically, we added a new composite figure in the main text and a new subsection of results. We started from the premise that, to be fair to the laissez-faire approach, we should let individuals internalize their own risk (ignoring the risk they pose to others if they become infected). To this end, we considered a theoretical scenario with age-specific fear of infection. In this scenario, households are grouped in three age classes, and each class has a level of fear that is proportional to the death risk in that class, in a way that aggregate fear is the same as

in the scenario in which fear is uniform across age classes. As we describe in the new subsection, this scenario does not show a departure from the results obtained previously with the homogeneous fear of infection, strengthening the similarity between the effects of behavioral response and NPIs on public health and epidemic outcomes.

In general, although it was possible to anticipate some qualitative results regarding the fear of infection, we believe that a quantitative evaluation was still lacking. We believe that our modeling analysis, based on a model coupling detailed economic and demographic information with data-driven contact networks provides relevant insights on one of the crucial debates that was spurred by the COVID-19 pandemic.

Comment R2.5: *My take on the main policy implications of the paper are the following: i. regardless the “fear of infection parameter”, the timing of intervention is better earlier than later, ii. The policy intervention should impose the closure only of customer facing industries. I find these results interesting, but a bit underwhelming with respect to my expectations while I was reading the description of the model. The authors claim that the model is “mostly applicable to the short-term management of emerging/re-emerging infectious diseases”. This is true, but it does not provide any novel management method. Most NPIs after the first wave of COVID-19 did in fact close only customer-facing industries, and tried to act timely to curb the re-emerging of the disease. So, the strategies suggested by the paper were already adopted during the COVID pandemic (if not during the first wave, during the following ones). Clearly, the model gives a more formal base for such interventions, but yet, it does not provide any new management tool and I don’t think it exploits fully the capabilities of the model.*

Answer R2.5: On one hand, we believe that giving a more formal and quantitative basis to the interventions that were adopted in later waves is by itself a worthwhile result. Before our paper, there were no theoretical models grounded on detailed real-world data that could address the joint epidemic-economic effects of different policies across socio-economic groups, and we think that quantitatively testing these policies is an important contribution. For instance, as the referee notes, “most” NPIs did restrict customer-facing industries, but some governments decided to support customer-facing industries during summer 2020 (see e.g. the eat-out-to-help-out scheme in the UK: <https://doi.org/10.1093/ej/ueab074>). This shows the need for continued publication of rigorous scientific evaluation of these policies.

On the other hand we also found not so intuitive results such as the fact that implementing NPIs late leads to almost no decrease in unemployment as providing relevant information for epidemic management. Another relevant finding for pandemic management is indeed the fact that while NPIs and spontaneous behavioral adaptation have similar effects on public health and economic outcomes, only NPIs can be implemented as soon as needed for highest effectiveness.

We now discuss these points in the manuscript.

Comment R2.6: *I suggest the authors to explore the following alternatives: i. Given the detailed pandemic module, would it be possible to build a tool capable of driving the NPIs in a more detailed real-time way? Assume that you had real-time (or almost real-time) infections in the Metropolitan Area. Using your model and the data-driven contact network, you could try to forecast short-run spreading of the disease and design NPIs to try to contrast such spreading. The pandemic module can provide the forecasts on the spreading of the disease, while the economic model provides the cost of curbing the pandemic. iii. In terms of different policies, the authors only analyse industry specific closures and timing of intervention in terms of date. If the aim is to provide a more general tool to respond to possible future pandemics, other important issues are to determine “when” to impose NPIs in terms of spreading of the disease (at what point of the diffusion dynamics should the NPIs be implemented?), and “for how long” (when*

should the NPIs be lifted?).

Answer R2.6: We thank the reviewer for suggesting us to explore optimized policies and strategies for pandemic management. We think that it is a good idea to showcase that our model can be used to study these policies, and have added Supplementary Section S6.6 to discuss the real-time activation and deactivation of protective measures. We assume that the policy maker has access to the model, as suggested by the reviewer, and can use the precise information coming from the model to close customer-facing industries when infections go beyond a given threshold, and then to reopen these industries when infections go back below the same or another threshold. By exploring these strategies we find interesting results, such as the possibility of quasi steady states with intermittent closures or the existence of multiple waves, depending on activation and deactivation thresholds. However, we also find that these policies are not more effective than the main policies that we considered (fixed start and end of protective measures) at improving epidemic and economic outcomes. We view this as a preliminary insight that can be gained by the model. However, we also think that designing realistic and feasible policies depends on assessing their practical feasibility and a case-by-case examination, based on resources, logistics as well as the objective functions to optimize, which is beyond this paper's scope.

Comment R2.7: *ii. Instead of analysing death, wouldn't it be relevant to analyse the hospital occupancy rate? This would provide a much more direct way to evaluate different policies. The question would be: given available resources, i.e., number of ICUs, what policies minimize unemployment without overrunning the health system.*

Answer R2.7: We agree with the reviewer that analyzing ICU occupancy could provide useful insights. We added Supplementary Section S6.4 to discuss this issue. We obtained the nominal ICU capacity in New York from official data, and assumed that going above 50% of the nominal capacity would cause excessive strain on the healthcare system. This is because the typical occupancy rate for ICU beds hovers between 80% and 90% throughout the healthcare system. This occupancy pattern indicates that any surge in COVID-19 ICU patients, exceeding 20% to 30% of the nominal capacity, could trigger a bed shortage. As a consequence, mechanisms to increase surge capacity would be activated, and policies designed to limit new admissions, such as the postponement of elective surgeries, would be implemented. On the other hand, given that the ICU capacity substantially increased during the first wave, the healthcare system could adapt to face emergency situations. Balancing these two effects, we judged that 50% of the nominal capacity was a reasonable threshold to be used as a benchmark.

With this threshold, we find that the only way to prevent putting an excessive burden on the healthcare system is to start protective measures early. Doing so, it may also not be necessary to close industries that do not involve customer contact, such as manufacturing and construction. As we know from Figure 3, this substantially reduces unemployment.

We view this as a preliminary exploration. Indeed, a discussion of ICU occupancy requires several assumptions on the management of patients, and the ramping up of hospital capacity, that have been rapidly changing during the early months of the pandemic and then during the course of the following years. Our preliminary analysis demonstrates that our model can be easily adapted to study these specific settings.

Comment R2.8: *In Section S4.3 the authors describe the estimation technique used to select the value of a subset of parameters. There is technical error in the estimation method. In fact, the authors state that "We run each parameter combination with a different random seed". Using a different seed for each parameter combination is increasing the estimation error. The correct estimation method uses the same random sequence for each parameter combination. The authors should use S simulations (as many as possible, taking into account computational resources) for each parameter combination using the same S seeds for each parameter*

combination.

Answer R2.8: We re-ran the calibration procedure following the suggestion by the reviewer. Instead of running 100,000 parameter combinations with 1 random seed, we now run 10,000 parameter combinations with 10 random seeds each. We also re-ran all the analyses and replaced all the figures in the main text and in the supplementary information. Although we observe very small differences with respect to the originally submitted manuscript, the obtained conclusions are unchanged.

Comment R2.9: *I didn't manage to run the code. If the paper is aimed at a wider audience, the code should be easier to run, or a more detailed readme should be provided.*

Answer R2.9: We apologize for the lack of detail. We have now provided a Docker image of the repository that should be easier to run. Code documentation has been improved as well.

3 Reviewer 3

Comment R3.1: *This is a highly innovative and interdisciplinary paper addressing a theoretically deep and crucially important problem: The health-economy tradeoff in mitigating pandemic disease. The novel methods developed to address this—using coupled economic and epidemic modules—are quite general and will be useful in future crises. For this paper, the model is calibrated to the COVID-19 pandemic with high quality data from the greater New York metropolitan area. Not only does it reproduce a number of crucial economic and epidemic dynamics, but the calibrated model is then used to study several core scenarios and their impacts on disparate industries, socioeconomic groups, and geographical areas. It represents a very powerful application of high-resolution data-driven heterogeneous agent modeling to a complex—and highly politicized—problem. As such, it advances the science and disciplines the policy debate in this area. Classical epidemic models ignore behavioral adaptation. There is now a recognition of its centrality and a powerful call for its inclusion, as in Nature Human Behavior (Bedson et al, 2022). That paper (on which I am a co-author) sounded the tocsin for better behavioral-epidemic modeling. This paper answers that call, advancing the field and fitting well (in my view) into NHB's emphasis on behavioral epidemiology. I have published widely on the need to include such cognitive drivers of behavior as contagious fear, which the authors represent in a convincing and transparent manner. The data quality is high, and it strongly supports the conclusions. The results are highly significant, specifically regarding equity dimensions of non-pharmaceutical interventions such as workplace closures. The references are thorough and literate on both the economic and epidemic dimensions of the problem. The writing is very clear throughout. What limitations I would note are noted by the authors themselves and certainly should not delay publication.*

Answer R3.1: We thank the reviewer for very positive evaluation of our manuscript and for bringing the Bedson et al. paper to our attention. We included it in our reference list, and mention it at the end of the discussion where we summarize which literature this paper can contribute to advance.

Comment R3.2: *I have one suggestion to consider. The word "price" does not occur in the paper, nor do such related Economics terms as the "price elasticity of demand" or "income elasticity of demand." For example, the authors focus on reduction in consumer demand due to*

fear, which is a central connection. To an economist, however, such effects as reductions in demand are mediated by prices. For example, COVID supply-chain disruptions reduced supplies, raising prices, which is what reduced consumer demand. I think it might strengthen the paper, and broaden its audience, to say simply that they recognize the role of price dynamics, but that they are not modeled explicitly here, but are understood to operating 'under the hood,' as it were. Perhaps include a reference or two to standard economic modeling of price effects during COVID-19, just to demonstrate the literacy I know to be present for some of the authors who—full disclosure—are colleagues.

Answer R3.2: The reviewer brings up an important point, that a subset of the authors acknowledge in another publication (ref. 17). It is definitely true that, in many cases of economic response to natural or human-made disasters that reduce supply, increases in prices mediate reductions in demand. This is what we saw, for instance, during the Russian invasion of Ukraine, when the price of Liquefied Natural Gas increased a lot, leading consumers and businesses in Europe to save on gas consumption and thus reduce demand, matching the reduced supply. However, we do not think that the price channel had any major role in the first wave of the COVID-19 pandemic, except for a few items that are unlikely to have macroeconomic impact (masks, sanitizers, etc). This is corroborated by macroeconomic statistics. For instance, checking the annualized percentage change in prices between Q2-2020 and Q1-2020 in NIPA table 1.1.7 (<https://apps.bea.gov/iTable/?reqid=19&step=3&isuri=1&1921=survey&1903=11>, accessed Sunday, April 23, 2023), we see that at the aggregate level it was the smallest change in the last three years.

More in general, while general equilibrium models used to estimate the economic impact of the COVID-19 pandemic (such as ref. 11 in the main text) mediate changes in demand through prices, dynamic input-output models from the natural disasters literature (such as refs 24-25 in the main text) do not. These papers focus on a very short period, during which prices have little time to change and rationing is more likely to play a first-order effect.

We now clarify why we do not consider price changes with a paragraph in Materials and Methods.

Comment R3.3: *Overall, this is an extremely innovative paper on a crucially important topic and certainly should be published. Its focus on behavior makes it especially suitable for Nature Human Behavior and I sincerely hope to see it in print.*

Answer R3.3: We thank again the reviewer for the positive evaluation of our work.

Decision Letter, second revision:

23rd August 2023

Dear Dr. Pangallo,

Thank you for your patience as we've prepared the guidelines for final submission of your Nature Human Behaviour manuscript, "The unequal effects of the health-economy tradeoff during the COVID-19 pandemic" (NATHUMBEHAV-22123224B). Please carefully follow the step-by-step instructions provided in the attached file, and add a response in each row of the table to indicate the changes that you have made. Please also address the additional marked-up edits we have proposed within the reporting

summary. Ensuring that each point is addressed will help to ensure that your revised manuscript can be swiftly handed over to our production team.

We would hope to receive your revised paper, with all of the requested files and forms within two-three weeks. Please get in contact with us if you anticipate delays.

Nature Human Behaviour offers a Transparent Peer Review option for new original research manuscripts submitted after December 1st, 2019. As part of this initiative, we encourage our authors to support increased transparency into the peer review process by agreeing to have the reviewer comments, author rebuttal letters, and editorial decision letters published as a Supplementary item. When you submit your final files please clearly state in your cover letter whether or not you would like to participate in this initiative. Please note that failure to state your preference will result in delays in accepting your manuscript for publication.

In recognition of the time and expertise our reviewers provide to Nature Human Behaviour's editorial process, we would like to formally acknowledge their contribution to the external peer review of your manuscript entitled "The unequal effects of the health-economy tradeoff during the COVID-19 pandemic". For those reviewers who give their assent, we will be publishing their names alongside the published article.

Cover suggestions

As you prepare your final files we encourage you to consider whether you have any images or illustrations that may be appropriate for use on the cover of Nature Human Behaviour.

We accept TIFF, JPEG, PNG or PSD file formats (a layered PSD file would be ideal), and the image should

be at least 300ppi resolution (preferably 600-1200 ppi), in CMYK colour mode.

ORCID

Non-corresponding authors do not have to link their ORCIDs but are encouraged to do so. Please note that it will not be possible to add/modify ORCIDs at proof. Thus, please let your co-authors know that if they wish to have their ORCID added to the paper they must follow the procedure described in the following link prior to acceptance:

Nature Human Behaviour has now transitioned to a unified Rights Collection system which will allow our Author Services team to quickly and easily collect the rights and permissions required to publish your work. Approximately 10 days after your paper is formally accepted, you will receive an email in providing you with a link to complete the grant of rights. If your paper is eligible for Open Access, our Author Services team will also be in touch regarding any additional information that may be required to arrange payment for your article.

Please note that *Nature Human Behaviour* is a Transformative Journal (TJ). Authors may publish their research with us through the traditional subscription access route or make their paper immediately open access through payment of an article-processing charge (APC). Authors will not be required to make a final decision about access to their article until it has been accepted. Find out more about Transformative Journals

For information regarding our different publishing models please see our Transformative Journals page. If

you have any questions about costs, Open Access requirements, or our legal forms, please contact ASJournals@springernature.com.

[REDACTED]

Best regards,
Alex McKay
Editorial Assistant
Nature Human Behaviour

On behalf of

Arunas Radzvilavicius, PhD
Senior Editor, Nature Human Behaviour
Nature Research

Reviewer #1:

Remarks to the Author:

Dear editors and Authors,

I carefully read the revised manuscript and I am more than happy f the changes they have made.

As per my opinion, the revised ms can be immediately published.

Kind Regards,

An Anonymous Referee

Reviewer #2:

Remarks to the Author:

I am satisfied with the answers of the authors. I have only a minor point:

Comment R2.8: I am not sure I fully understand the algorithm you use to calibrate the model. In the original version of the paper, as far as I understand, you used a different random seed for each combination of parameters.

Now you state that you use more random seed, and less parameter combination. Are you using the same

10 random seeds for each combination of parameters? The point was not about increasing the number of seeds (which can still be useful if the model displays some non-ergodic properties), but to keep the random seed (or seeds, if more than one) fixed.

In the text you now write: “We run each parameter combination with 10 different random seeds”. You should keep the random seeds fixed as you change the parameters values. Are you doing so? If yes, could you clarify this in the text?

Author Rebuttal, second revision:

1 Reviewer 1

Comment R1.1: *Dear editors and Authors, I carefully read the revised manuscript and I am more than happy of the changes they have made. As per my opinion, the revised ms can be immediately published. Kind Regards, An Anonymous Referee*

Answer R1.1: We thank the reviewer for their time evaluating our work and providing suggestions.

2 Reviewer 2

Comment R2.1: *I am satisfied with the answers of the authors. I have only a minor point: Comment R2.8: I am not sure I fully understand the algorithm you use to calibrate the model. In the original version of the paper, as far as I understand, you used a different random seed for each combination of parameters. Now you state that you use more random seed, and less parameter combination. Are you using the same 10 random seeds for each combination of parameters? The point was not about increasing the number of seeds (which can still be useful if the model displays some non-ergodic properties), but to keep the random seed (or seeds, if more than one) fixed. In the text you now write: “We run each parameter combination with 10 different random seeds”. You should keep the random seeds fixed as you change the parameters values. Are you doing so? If yes, could you clarify this in the text?*

Answer R2.1: Yes, we are using the same 10 random seeds for each combination of parameters. We now clarify this in the text. The new sentence reads: “We determine a set of 10 random seeds, and run each parameter combination using these seeds, as this is known to reduce the estimation error of ABC.”

Thank you for the detailed feedback and suggestions, we think they greatly helped improve the quality of the paper.

Final Decision Letter:

Message: Dear Dr Pangallo,

We are pleased to inform you that your Article "The unequal effects of the health-economy tradeoff during the COVID-19 pandemic", has now been accepted for publication in Nature Human Behaviour.

Please note that *Nature Human Behaviour* is a Transformative Journal (TJ). Authors may publish their research with us through the traditional subscription access route or make their paper immediately open access through payment of an article-processing charge (APC). Authors will not be required to make a final decision about access to their article until it has been accepted. [Find out more about Transformative Journals](https://www.springernature.com/gp/open-research/transformative-journals)

Authors may need to take specific actions to achieve [compliance with funder and institutional open access mandates](https://www.springernature.com/gp/open-research/funding/policy-compliance-faqs). If your research is supported by a funder that requires immediate open access (e.g. according to [Plan S principles](https://www.springernature.com/gp/open-research/plan-s-compliance)) then you should select the gold OA route, and we will direct you to the compliant route where possible. For authors selecting the subscription publication route, the journal's standard licensing terms will need to be accepted, including [self-archiving policies](https://www.springernature.com/gp/open-research/policies/journal-policies). Those licensing terms will supersede any other terms that the author or any third party may assert apply to any version of the manuscript.

With best regards,

[redacted]